# Continual Learning Based on Sub-Networks and Task Similarity

## Abstract

Continual learning (CL) has two main objectives: preventing *catastrophic forgetting* (CF) and encouraging *knowledge transfer* (KT) across tasks. The existing literature mainly tries to overcome CF. Although some papers have focused on both CF and KT, they may still suffer from CF because of their ineffective handling of previous tasks and/or poor task similarity detection mechanisms to achieve KT. This work presents a new CL method that addresses the above issues. First, it overcomes CF by isolating the knowledge of each task via a learned mask that indicates a sub-network. Second, it proposes a novel technique to compute how important each mask is to the new task, which indicates how the new task is similar to an underlying old task. Similar tasks can share the same mask/sub-network for KT, while dissimilar tasks use different masks/sub-networks for CF prevention. Comprehensive experiments have been conducted using a range of NLP problems, including classification, generation, and extraction to show that the proposed method consistently outperforms prior state-of-the-art baselines.[1]

## 1 Introduction

This paper studies continual learning (CL) of a sequence of natural language processing (NLP) tasks in the task continual learning (Task-CL) setting. It deals with both *catastrophic forgetting* (CF) (McCloskey & Cohen, 1989) and *knowledge transfer* (KT) across tasks. In Task-CL, the task ID is provided for each test case in testing. In learning, after a task is learned, its data is no longer accessible. Another CL setting is class continual learning (Class-CL), which provides no task ID in testing and it solves a different type of problems.

Existing research in CL has almost exclusively focused on overcoming CF (Kirkpatrick et al., 2016; Serrà et al., 2018; Wortsman et al., 2020). Limited work has being done on KT except (Ke et al., 2020; 2021; Wang et al., 2022). But KT is particularly important for NLP because many tasks in NLP share similar knowledge that can be leveraged to achieve better accuracy. We humans are also particularly good at leveraging prior knowledge to help learn new skills. To achieve KT in learning a new task, CAT (Ke et al., 2020) first detects previous tasks that are similar to the current task so that the current task learning can leverage the knowledge learned from the similar past tasks. CAT uses the hard-attention mechanism in HAT (Serrà et al., 2018) to deal with CF, which masks out those important neurons for each task so that the training of new tasks cannot change them in back-propagation. However, different tasks can share neurons. This approach has a major problem for KT that is very hard, if not impossible, to solve. After the similar previous tasks are detected, CAT opens the masks of these tasks so that the new task learning can modify their parameters to achieve both forward and backward KT. This clearly helps KT. But this can cause CF for dissimilar tasks that share parameters with those similar tasks. CAT's task similarity comparison method based on the transfer learning performance can be quite inaccurate too. The KT methods in (Ke et al., 2021; Wang et al., 2022) are based on instance-level feature similarity comparison using dot product or cosine, which can be inaccurate as well (see experiment results in Sec. 4.2).

To deal with these issues, we would like to have (1) a learning method that can isolate the knowledge of each task without parameter overlapping among tasks to deal with harmful interference in KT and (2) a task similarity detection method that is directly related to the loss of previous tasks (even though we do not have their data) for more accurate similar task detection.

---

[1]The code has been uploaded as supplementary materials.

For (1), we draw inspiration from the sub-network masking idea in (Wortsman et al., 2020), where the underlying backbone network is fixed but a binary mask is learned to find a sub-network for each task, which encodes the model for the task. The mask is basically a set of binary gates that indicates which parameters in the backbone network should be used for a task model. Thus, different task models have no interference to cause CF although sub-networks of multiple tasks can share neurons and parameters because the underlying backbone network is fixed and shared by all tasks. Although this helps learn a sub-network to achieve no interference (CF) in transfer, the original (Wortsman et al., 2020), by design, cannot do KT. It is still very challenging to detect task similarity and to know what level of similarity is similar enough to ensure positive transfer. If a wrong similarity threshold is used, CF will be serious. To this end, we propose a novel method to detect similarity in (2).

For (2), we propose to determine whether a previous task $k$ is similar to the current task $t$ by assessing the importance of the mask (which represents the model or sub-network of a task) for the previous task $k$ to the current task $t$. To compute the importance score, we make use of an effective idea from the network pruning community (Michel et al., 2019). In Michel et al. (2019), the gradient on each parameter is serving as the importance of the parameter. The less important parameters (determined by a threshold) are regarded as unimportant and removed to reduce the network size. However, it is not obvious how we can compute the importance of each mask with its sub-network to the current task and how to determine whether the current task is similar enough to a previous task based on the importance so that they can perform KT. This paper proposes a novel method to perform the above two functions. A set of virtual/dummy gate variables are introduced to represent the mask/sub-network so that we can compute the gradient of each mask/sub-network. The gradient, based on the current task data and directly related to current task loss, serves as the importance of the mask/sub-network to the current task. The more important a mask/sub-network is, it is more likely that the previous task that has used the mask is similar to the current task. To mitigate the possible forgetting, a novel importance comparison mechanism is also proposed to take the previous task gradient into account.

Based on the proposed idea, a new method, called **TST** (**T**ask-CL based on **S**ub-networks and **T**ask similarity), is proposed. TST is evaluated using datasets for classification, generation, and extraction with similar tasks and dissimilar tasks. The results demonstrate the high effectiveness of TST.

In summary, this paper makes two key contributions.

1. It proposes a new Task-CL method TST based on sub-networks and task similarity. TST not only overcomes CF but also enables effective KT. For KT, it learns the current task in the sub-network of a previous task without interference with any other tasks and thus will not cause any CF for the other tasks. This cannot be achieved by other existing methods.

2. It proposes a novel task similarity detection method based on gradients computed on masks. This method is simple and yet highly effective. It is instrumental for effective KT.

## 2    RELATED WORK

**Continual learning.** Existing CL work mainly focused on overcoming CF: (1) *Regularization-based approaches* (Kirkpatrick et al., 2016; Lee et al.; Seff et al., 2017; Zenke et al., 2017; Rusu et al., 2016) add a regularization in the loss to penalize changes to parameters that are important to previous tasks. (2) *Gradient projection* (Zeng et al., 2019) ensures the gradient updates occur in the orthogonal direction to the input of old tasks. (3) *Parameter isolation* (Serrà et al., 2018; Ke et al., 2020; Mallya & Lazebnik, 2018; Fernando et al., 2017; Wortsman et al., 2020) learns a dedicated sub-network for each task and masks it out in learning new tasks. (3) *Replay-based approaches* (Rebuffi et al., 2017; Lopez-Paz & Ranzato, 2017; Chaudhry et al., 2019; Wang et al., 2020), retain some training data of old tasks and use them in learning a new task. The methods in (Shin et al., 2017; Kamra et al., 2017; Rostami et al., 2019; He & Jaeger, 2018) learn data generators and generate old task data for learning a new task. These are clearly very different from our method as we don't use any replay data.

**Continual learning in NLP.** Above existing approaches usually do not use a pre-trained model. However, in NLP, almost all recent CL/non-CL techniques use pre-trained language models (LMs). We categorize them into 3 types based on which part in the pre-trained LM is trainable. The first one below belongs to replay or regularization families and last two types belong to parameter-isolation. (1). *Transformer-updating based* - this family updates the Transformer directly. IDBR (Huang et al., 2021) disentangles task-shared and task-specific knowledge in BERT via regularizations. LAMOL (Sun

et al., 2020) is a replay method based on GPT-2. It generates previous task samples before learning the new task. (2). *Prompt-based* - a prompt (Lester et al., 2021) is an added structure to the input layer. C-PT (Zhu et al., 2022) learns a separate prompt for each task. L2P (Wang et al., 2022) learns a prompt pool for all tasks and prevents CF by choosing the matched prompt and allowing transfer by sharing the same prompt pool. (3). *Adapter-based* - an adapter (Houlsby et al., 2019) is an added structure in each layer of the Transformer. AdapterCL (Madotto et al., 2020) trains a separate adapter for each task. It thus has no CF or KT. CTR (Ke et al., 2021) trains a shared adapter for all tasks. It prevents CF via task masks (Serrà et al., 2018) and achieves KT via capsule networks (Sabour et al., 2017). Our method TST also belongs to this family but TST updates the mask instead of the adapter. It needs no replay samples or regularization and can effectively detect the task similarity based on the mask importance. We have discussed the advantages of TST in Sec. 1.

**Network pruning as importance computation.** It is known that many parameters in a neural network are redundant and can be pruned (Li et al., 2021; Lai et al., 2021). This has also been shown for pre-trained Transformer (Chen et al., 2020; Lin et al., 2020; Gao et al., 2021; Michel et al., 2019; Voita et al., 2019). One method is to discard the parameters with small absolute values (Han et al., 2015; Guo et al., 2016). Other methods prune at higher levels. In a Transformer-based model, these include pruning the attention head (Michel et al., 2019; Voita et al., 2019; McCarley et al., 2019) and pruning sub-layers in a standard Transformer layer (Fan et al., 2020; Sajjad et al., 2020). However, the above methods are not directly applicable to us as we need to compute the importance of mask to detect task similarity, while the above approaches are all for the importance of network parameters.

## 3 PROPOSED TST TECHNIQUE

TST is designed for both CF prevention and KT (knowledge transfer). Most of existing methods focus on CF prevention. The few methods that also do KT are based on HAT, which, as we discussed in Sec. 1, *cannot* completely isolates the parameters for different tasks and the shared parameters still cause forgetting when updating the sub-networks for similar tasks. This work borrows the idea of *sub-network masking* from (Wortsman et al., 2020), which trains a mask to isolate a sub-network for a task with the network parameters fixed. Thus, different tasks have no interference to cause CF.

The proposed method maintains a a shared mask pool. In learning a new task, the system first computes the importance of each mask in the pool to the new task based on the gradient of a virtual gate variable representing the mask (Sec. 3.2). Based on the mask importance, if the mask of a previous task is likely to improve both the previous task and the current task, the mask and its underlying sub-network will be used to train the current new task for KT (Sec. 3.3). If such a mask is not found, the unused mask in the pool will be trained to find a new sub-network for the task so that it will not cause CF for any previous task. The whole process is illustrated in Figure 1, which is based on the adapter in a pre-trained Transformer framework (Houlsby et al., 2019)[2]. Both the adapter (randomly initialized) and the pre-trained Transformer are fixed. Only the mask, which is applied only to the adapter, is trained to find a sub-network in the adapter for each task.

### 3.1 SUB-NETWORK MASKING AND MASK POOL

Let the *mask pool* with $K$ masks be $\{M^{(k)}\}_{k=1}^{K}$ ($K$ is a hyper-parameter). To learn a task, a mask is selected from the pool and trained. Each mask $M^{(k)}$ is of the same size as the parameter set ($W$) of the inserted adapter. The detailed mask training is given in Appendix B. A mask is always binary and indicates a sub-network by element-wise multiplying with the adapter parameters,

$$o^{(k)} = f(x, W \otimes M^{(k)}), \tag{1}$$

where $\otimes$ is element-wise multiplication, $f$ refers to the adapter. $W$ is fixed and shared by all tasks. Only $M^{(k)}$ is trainable. $W \otimes M^{(k)}$ represents a sub-network and can be indicated by mask $M^{(k)}$.

### 3.2 COMPUTING MASK IMPORTANCE

Since the network weights are always fixed, a simple way to perform CL is to use a different mask for each task so that the resulting sub-networks for different tasks will not affect each other. This

---

[2]Details of adapter is given in Appendix A.

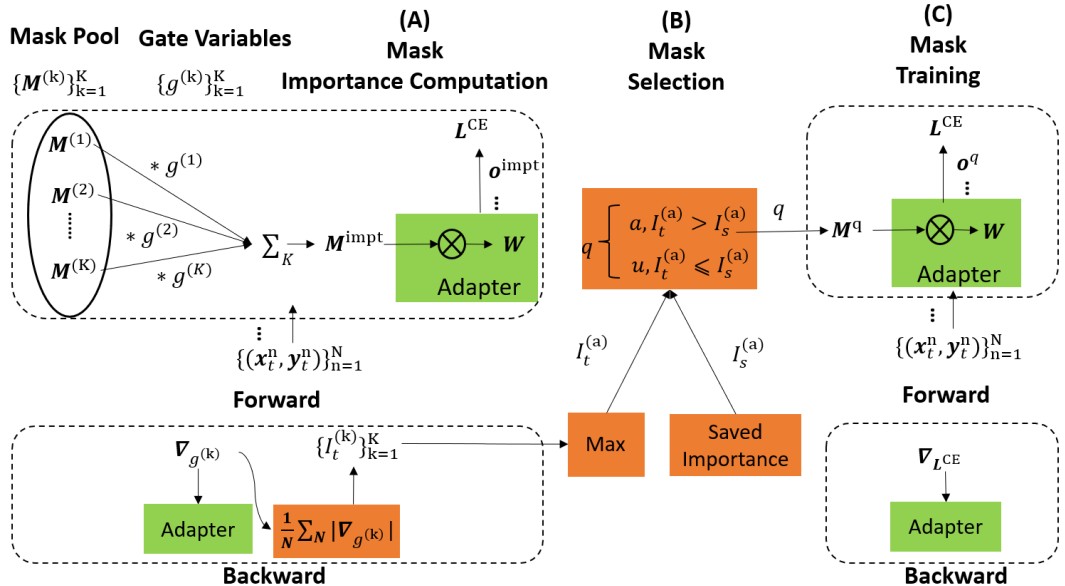

Figure 1: Illustration of TST (from left to right). There are $K$ masks in the mask pool, where each mask is of the same size as the adapter weight and always binary (Appendix B) before learning. When a new task $t$ comes, TST first multiplies each mask with an virtual gate variable (with value of 1). The resulting gated masks are then combined via summation to produce the combined mask for importance computation (Eq. 2). **(A)** shows the mask importance computation (Sec. 3.2). We mask the adapter using the combined mask $M^{\text{impt}}$ and compute the gradient of each virtual gate variable via cross-entropy loss ($\mathcal{L}^{\text{CE}}$). The normalized magnitude of the resulting gradient serves as the importance of each mask to the current task $t$ (Eqs. 4 and 5). **(B)** shows the mask selection (Sec. 3.3). We take the mask with the highest importance as candidate mask $a$ and select a mask using the saved importance of the previous task that used the mask. Only the mask of higher importance for the current task data is selected. Otherwise, we select an unused mask $u$. (Eqs. 6 and 7) **(C)** shows the mask training (Sec. 3.4). Using the selected mask $q$, we can mask the adapter and train the mask with cross-entropy. As a result, we share the same mask for similar tasks and isolate the mask from dissimilar tasks, achieving both forgetting prevention and knowledge transfer (KT).

gives us a chance to safely update the masks of sub-networks of similar tasks without affecting any other tasks. However, this cannot achieve KT (knowledge transfer), which is a main goal of this paper. Thus, before learning a new task, we propose to select a mask for KT based on task similarity, where each task may use the same mask as a previous task (for KT) or a unused mask in the pool. The intuition is that if two tasks are similar, they can share the *same* mask from the pool to encourage knowledge transfer; otherwise, the new task should select a unused mask to avoid forgetting. We present how to compute the task similarity next.

**Virtual gate variables.** Intuitively, if a mask is important for two different tasks, the two tasks are similar. This motivates us to compute the importance of each mask in the pool to the current task based on its data. To this end, we introduce a set of *gate variables*, $\{g^{(k)}\}_{k=1}^{K}$, where each variable corresponds to a mask in the pool. The variable serves as a proxy for computing the gradient on each mask, which is used to compute the importance of the mask to the current task. We call these *virtual* gate variables since each $g^{(k)}$ is initialized to 1, and is not changed. This is because we only need its gradient to compute the importance of a mask and will not use the gradient to update any parameter. We first define the virtual network of masks for our importance computation (see Figure 1(A)),

$$M^{\text{impt}} = \sum_{k=1}^{K} (g^{(k)} * M^{(k)}),$$ (2)

where $g^{(k)}$ is a scalar-valued gate variable and $M^{(k)}$ is a mask. Eq. 2 basically combines all masks ($g^{(k)} * M^{(k)}$) so that we can compute the gradient on each gate variable based on the current task data. The resulting combined mask, $M^{\text{impt}}$, is then used to mask the adapter, similar to Eq. 1

$$o^{\text{impt}} = f(x, W \otimes M^{\text{impt}}),$$ (3)

$o^{\text{impt}}$ can be used to compute the loss based on the current task data and the gradient on each gate variable $g^{(k)}$ gives us the importance of the corresponding mask $\boldsymbol{M}^{(k)}$.

**Mask importance based on the gradients of gate variables.** To compute the mask importance, we borrow the idea from neural network punning (Michel et al., 2019). Given the data from the current task $t$, $D_t = \{(\boldsymbol{x}_t^n, \boldsymbol{y}_t^n)\}_{n=1}^N$, the importance of a mask is estimated with a gradient-based score.

$$I_t^{(k)} = \frac{1}{N} \sum_{n=1}^{N} |\nabla_{g^{(k)}}^n|, \tag{4}$$

where $\nabla_{g^{(k)}}$ is the gradient on the gate variable $g^{(k)}$,

$$\nabla_{g^{(k)}}^n = \frac{\partial \mathcal{L}^{\text{impt}}(\boldsymbol{x}_t^n, \boldsymbol{y}_t^n)}{\partial_{g^{(k)}}}, \tag{5}$$

where $\mathcal{L}^{\text{impt}}$ (*impt* stands for *importance*) is a problem-specific loss function[3]. Again, the gradient in Eq. 5 is only for importance computation and is not used to update the parameters. Only the final selected mask is trained in Sec. 3.4.

**$\mathcal{L}^{\text{impt}}$ for mask importance.** Eq. 4 offers a way to compute the mask importance w.r.t. a given loss $\mathcal{L}^{\text{impt}}$. A nature idea is to use the cross-entropy loss[4], $\mathcal{L}^{\text{CE}}$, as $\mathcal{L}^{\text{impt}}$. By doing so, we can easily get the importance scores of all masks in the mask pool to the current task[5]. Note that the resulting importance score $I_t^{(k)}$ is saved for each task.

### 3.3 Selecting a Mask to Learn Task $t$ based on Mask Importance

Recall that the goal of computing mask importance is to find the task similarity and select a mask for the current task based on it. To achieve this, we propose a 2-step mechanism:

**(1) Find the mask with highest importance as the candidate for knowledge transfer.** We leverage the importance scores computed using the data of the current task $t$, $\{I_t^{(k)}\}_{k=1}^K$. Intuitively, the highest score indicates the most useful sub-network for task $t$. If this sub-network has been used by a previous task, it indicates the previous task's sub-network is helpful to task $t$ and the two tasks are very likely to be similar. Formally, in learning task $t$, we first get the highest importance to task $t$,

$$I_t^{(a)} = \max(\{I_t^{(1)} ... I_t^{(K)}\}). \tag{6}$$

where $a$ is the ID of the most useful/important mask (sub-network) to the current task $t$. The previous task that has used $\boldsymbol{M}^{(a)}$, denoted as task $s$[6], is then the candidate similar previous task to task $t$.

**(2) Select a mask for task $t$.** We cannot simply say that the previous task $s$ is similar to current task $t$. We want to ensure that the update of the mask for $s$ by $t$ will not cause forgetting for $s$, and can result in improvement for both. Eq. 6 is only about whether a previous used sub-network $\boldsymbol{M}^{(a)}$ can help task $t$, it does not consider whether the updating of the $\boldsymbol{M}^{(a)}$ by $t$ will cause forgetting for $s$. Therefore, we need one more step to be confident that forgetting is unlikely to occur due to the updating of $\boldsymbol{M}^{(a)}$ by $t$'s data. Since task $s$'s data is no longer accessible, we propose to use the saved importance score ($I_s^{(a)}$) computed using $s$'s data for mask $a$ to achieve the goal[7]. This is because $I_s^{(a)}$ is computed from the previous task $s$'s data when we were in task $s$ and is directly related to the

---

[3]This paper focuses on supervised tasks. Future work will extent TST to unsupervised tasks like topic modeling (Gupta et al., 2020).

[4]Different problems in our experiments (generation, classification and extraction) all use cross-entropy loss.

[5]To facilitate the importance comparison, the importance scores for each mask $k$ is normalized so that the importance scores for all masks have a mean of 0 and standard deviation of 1. To simplify the notation, we use the same $I_t^{(k)}$ to represent the normalized importance of mask $\boldsymbol{M}^{(k)}$ computed from current task $t$.

[6]If multiple tasks have selected mask $a$, we use the task with the highest importance as task $s$. We realize the mask has been updated if it is shared by multiple tasks and the importance needs to be updated as well, which we do not do. Our current simple approach works well in our experiments. We leave the issue to our future work.

[7]$I_s^{(a)}$ is computed and saved *after* training task $s$ (to ease the notation, we still use the same $I_s^{(a)}$). This is to ensure that $I_s^{(a)}$ is computed based on the *trained* mask and is comparable to $I_t^{(a)}$.

previous task loss. For mask $a$, we now have two importance scores; $I_t^{(a)}$ computed from the current task $t$ and $I_s^{(a)}$ computed from the previous task $s$ which learned the mask $a$. We then compare their values and select the mask via,

$$q = \begin{cases} a, & I_t^{(a)} > I_s^{(a)} \\ u, & otherwise, \end{cases} \tag{7}$$

where $u$ refers to any mask in the pool that has not been selected and used by a previous task. The above inequality can tell us whether the updating of the mask $a$ by the current task is likely to help the previous task $s$. The rationale for Eq. 7 is that $I_s^{(a)}$ refers to the normalized magnitude of the gradient that can decrease the previous task loss most. Based on the concept of gradient-based importance in (Michel et al., 2019), since mask $a$ is more important to the current task $t$ than to the previous task $s$, the current task can potentially improve task $s$ and vice versa to achieve KT. It is thus reasonable to use the mask $a$ (and its associated sub-network) to also learn the current task $t$. If mask $a$ is not as important to the current task $t$ as to the previous task $s$, the current task data is unlikely to help the previous task $s$ and vice versa, and a unused mask in the pool should be used to learn $t$ to avoid CF on any previous task. Our empirical results show that Eq. 7 works well.

### 3.4 TRAINING THE SELECTED SUB-NETWORK

Using the selected mask $M^{(q)}$ for the current task, we can train the task by plugging $M^{(q)}$ in Eq. 1 with the cross-entropy loss $\mathcal{L}^{\text{CE}}$.

## 4 EXPERIMENTS

We now evaluate the proposed system TST. We first learn all tasks sequentially. After that, their task models are tested using their respective test sets. TST does not use any replay data.

### 4.1 DATASETS AND BASELINES

**Datasets:** We use the five datasets covering a wide range of NLP problems, including classification, generation, and extraction. We introduce each of them bellow. For detailed datasets statistics, please see Appendix C. **(1). ASC** (*Aspect Sentiment Classification*). This dataset is from (Ke et al., 2021) and has 19 tasks. Each task classifies the opinion (*positive*, *negative*, or *neutral*) in a review sentence at the aspect-level of a product or service. For example, "*The picture is good but the sound is poor*" about a TV expresses a *positive* opinion about the aspect "picture" and a *negative* opinion about the aspect "sound." **(2). CCD** (*Continual Classification Dataset*). This is a text classification dataset (de Masson d'Autume et al., 2019) that is popular in continual learning for NLP. It contains 5 tasks include AGNews (news classification), Yelp (sentiment analysis), Amazon (sentiment analysis), DBpedia (Wikipedia article classification) and Yahoo (questions and answers categorization).[8] **(3). SUM** (*ConvoSum*). This is a conversational abstractive summarization dataset with 6 tasks or domains (Fabbri et al., 2021). Given conversation from a domain, the system generates its summary. **(4). DRG** (*Dialogue Response Generation*). This is a popular task-oriented dialogue response dataset (Multi-WoZ2.0) (Ramadan et al., 2018) with 5 tasks/domains. Given the intent and dialogue state (slot-value pairs containing messages to express), the system is expected to generate a response. **(5). NER** (*Named Entity Recognition*). This data consists of 5 tasks, including **conll03** (Sang & Meulder, 2003), **wikigold** (Balasuriya et al., 2009), **btc** (Derczynski et al., 2016), **re3d** (Laboratory, 2017), and **gum** (Zeldes, 2017). Each task needs to classify mentions into pre-defined entity types. [9]

Among the 5 datasets, two of them (ASC and NER) have similar tasks and our goal is to achieve both CF prevention and KT. Three of them (SUM, CCD, DRG) consist of dissimilar tasks that have little shared knowledge to transfer. Then the main goal is to ensure there is little or no CF.

**Baselines.** We setup 12 baselines, including both *non-continual* and *continual learning* (CL) methods.

---

[8]Since each of these datasets is quite large, we randomly sampled 500 samples from each class for each task due to our resource limitations.

[9]Due to resource limitations, we randomly sampled 200 samples for each task.

***Non-CL baselines***: **Comb/MTL** and **Comb/MTL (Adapter)** [10] train tasks in a multi-task or data combined setting, where the former trains the whole LM and the latter trains only the adapter. These two are widely accepted as upper bounds of continual learning. **ONE** builds a separate model for each task by fine-tuning the LM, which clearly has no knowledge transfer (KT) or CF. **ONE (Adapter)** (Madotto et al., 2020) trains an adapter for each task separately (called AdapterCL in its original paper). **ONE (Prompt)** (Zhu et al., 2022) trains a prompt for each task (called C-PT in its original paper).

***CL baselines***. The CL setting includes an *naive continual learning* (**NCL**) method where the system learns the tasks one by one with no mechanism to deal with CF or to encourage transfer, and 7 state-of-the art *task continual learning* (Task-CL) methods.

The 7 CL baselines include: 4 adapter-based methods **CTR** (Ke et al., 2021), **HAT** (Serrà et al., 2018), **SupSup** (Wortsman et al., 2020) and **CAT** (Ke et al., 2020). They all train a shared adapter while SupSup trains a sub-network mask on the fixed adapter. HAT is one of the most effective Task-CL methods with little forgetting. **CTR** (Ke et al., 2021) encourages transfer via capsule networks and transfer routing. SupSup uses the similar sub-network masking method as TST but without any consideration on knowledge transfer. CAT and CTR are two systems that deal with both CF and KT. 1 prompt-based method **L2P** (Wang et al., 2022), which trains a prompt pool to transfer task knowledge and a key-value pair prompt selection strategy to select the task-specific prompt (it thus deals with both KT and CF); 2 baselines that modify the Transformer: **LAMOL** (Sun et al., 2020) is a pseudo-replay method using GPT-2. **EWC** (Kirkpatrick et al., 2016) is a regularization method.

## 4.2 Evaluation Results and Analysis

Since we need a backbone LM that can do both classification and generation, we adopt $BART_{LARGE}$ (Lewis et al., 2020) as our LM. Fine-tuning of BART follows the standard practice [11]. Due to space limits, detailed *hyperparameters* are given in Appendix D. Since the order of the tasks in a sequence may impact the final results, we ran 5 randomly sampled task sequences. We compute different metrics for different types of tasks using their standard metrics.[12] Table 1 gives the average result of each system on all tasks of each dataset over the 5 random task sequences. Note that LAMOL for NER is not included as it is not obvious how to adapt LAMOL for token-level classification.

**Superiority of TST.** Table 1 shows that TST clearly outperforms all baselines for tasks with shared knowledge (ASC and NER), and has no forgetting for tasks with little shared knowledge (SUM, CCD and DRG). Below, we discuss additional observations.

(1). TST achieves both CF prevention and knowledge transfer (KT). Using ONE as control, we can see TST outperforms ONE in two datasets (ASC and NER) with similar tasks, indicating effective KT. TST achieves similar results to ONE in other datasets, indicating effective CF prevention. We can also see TST is very similar to MTL/Comb. This again shows the effectiveness of TST.

(2). TST is more effective than the baseline CL systems (EWC, HAT, SupSup, LAMOL) that only deal with CF. This is because regularization-based EWC sacrifices accuracy for overcoming CF and parameter-isolation based SupSup prevents any possible knowledge transfer (KT) and is thus poorer in ASC and NER as these two datasets contain similar tasks. The other 3 datasets all consists of very dissimilar tasks with little shared knowledge to transfer, so TST is similar to the baselines that only deal with CF like SupSup. HAT has little KT in classification tasks, which makes ASC poorer. It has forgetting in generation tasks as it cannot isolate parameters in the shared LM head. LAMOL has only weak CF prevention and KT as it highly relies on the replay data, which could be of low quality.

(3). TST is also more effective than the baseline CL systems that try to deal with both CF and KT (CAT, CTR and L2P). Among these systems, L2P performs the worst due to the poor prompt selection

---

[10]For classification datasets (ASC, CCD and NER), we conduct a multi-task learning (MTL) experiment. For generation datasets (SUM and DRG), it is not possible to use MTL as the language modeling head on top of BART is a linear layer with weights tied to the input embeddings. We follow the standard practice (e.g., Qin & Joty (2022); Madotto et al. (2020)) and pool all data together to train a single shared head (called "Comb").

[11]For the ASC tasks, we adopt the ASC formulation in (Xu et al., 2019), where the aspect term and sentence are concatenated via ``. The opinion is predicted using the average over all tokens.

[12]Specifically, we use Macro-F1 and accuracy for the sequence-level classification tasks (ASC and CCD), where Macro-F1 (MF1) is the primary metric because highly imbalanced classes in ASC introduce biases in accuracy. We use Rouge score (R1, R2 and RL) for SUM, BLEU score for DRG and F1 for NER.

| Scenario | Data | SUM | | | ASC | | CCD | | DRG | NER |
|----------|------|-----|-----|-----|-----|-----|-----|-----|-----|-----|
| | Model | R1 | R2 | RL | MF1 | Acc | MF1 | Acc | BLEU | F1 |
| Non-CL | Comb/MTL | 39.3861 | 10.3258 | 35.0513 | 0.9228 | 0.9482 | 0.9057 | 0.9055 | 0.2529 | 0.6333 |
| | Comb/MTL (Adapter) | 38.8423 | 11.3795 | 34.8061 | 0.9217 | 0.9465 | 0.9109 | 0.9114 | 0.2450 | 0.6061 |
| | ONE | 39.0738 | 10.7076 | 35.2501 | 0.8555 | 0.9150 | 0.9107 | 0.9109 | 0.2414 | 0.5933 |
| | ONE (Adapter) | 38.9000 | 11.5449 | 35.2323 | 0.8395 | 0.9090 | 0.9078 | 0.9081 | 0.2342 | 0.5669 |
| | ONE (Prompt) | 30.6709 | 7.2285 | 27.5339 | 0.7646 | 0.8554 | 0.8623 | 0.8628 | 0.1267 | 0.4590 |
| CL | NCL | 32.6824 | 6.8418 | 29.1898 | 0.8925 | 0.9304 | 0.8508 | 0.8510 | 0.2231 | 0.4919 |
| | EWC | 32.6362 | 7.1160 | 29.0004 | 0.8836 | 0.9260 | 0.8727 | 0.8737 | 0.1830 | 0.5176 |
| | HAT | 37.1127 | 10.4009 | 33.5331 | 0.8933 | 0.9328 | 0.9021 | 0.9023 | 0.2147 | 0.5231 |
| | SupSup | 38.3660 | **11.6289** | 34.7497 | 0.8898 | 0.9335 | 0.9098 | 0.9100 | **0.2471** | 0.5893 |
| | LAMOL | 10.8817 | 1.3885 | 6.8711 | 0.8462 | 0.9017 | 0.5444 | 0.6704 | 0.1996 | — |
| | CAT | 37.2350 | 10.5304 | 33.7733 | 0.8431 | 0.8898 | 0.9082 | 0.9090 | 0.2172 | 0.5073 |
| | CTR | 37.3360 | 10.7305 | 33.6853 | 0.8886 | 0.9294 | 0.9054 | 0.9058 | 0.2139 | 0.5185 |
| | L2P | 26.6538 | 4.8220 | 23.8983 | 0.7481 | 0.8464 | 0.8535 | 0.8554 | 0.0852 | 0.4422 |
| | TST (forward) | 38.7075 | 11.3096 | 34.8344 | 0.9118 | 0.9457 | 0.9123 | 0.9125 | 0.2447 | 0.6159 |
| | TST | **38.7758** | 11.3652 | **34.9935** | **0.9161** | **0.9453** | **0.9104** | **0.9103** | 0.2425 | **0.6169** |

Table 1: Performance for different type of tasks, averaged over 5 random sequences (the standard deviation is reported in Appendix E. The results of individual sequences are reported in Appendix F). "—" means not applicable. TST (forward) is the average test performance of each task when it was first learned (refer to the text in Sec. 4.2). We bold the best performance within CL baselines.

| Scenario | Data | SUM | | | ASC | | CCD | | DRG | NER |
|----------|------|-----|-----|-----|-----|-----|-----|-----|-----|-----|
| | Model | R1 | R2 | RL | MF1 | Acc | MF1 | Acc | BLEU | F1 |
| CL | NCL | 8.3709 | 5.5409 | 5.9023 | -0.0044 | 0.0031 | 0.0567 | 0.0564 | 0.0301 | 0.1549 |
| | EWC | 2.5835 | 1.6784 | 2.0420 | 0.0781 | 0.0278 | 0.0894 | 0.0817 | 0.0078 | 0.0753 |
| | HAT | 1.0993 | 1.2171 | 0.7840 | -0.0088 | -0.0036 | -0.0007 | -0.0008 | 0.0148 | -0.0344 |
| | SupSup | 0.0000 | 0.0000 | 0.0000 | 0.0000 | 0.0000 | 0.0000 | 0.0000 | 0.0000 | 0.0000 |
| | LAMOL | 3.9017 | 1.4158 | 2.3189 | -0.0188 | -0.0152 | 0.3226 | 0.2730 | 0.0229 | — |
| | CAT | 1.8510 | 1.4146 | 1.3684 | 0.0205 | 0.0081 | -0.0013 | -0.0021 | 0.0226 | 0.0081 |
| | CTR | 1.4349 | 0.9002 | 0.8590 | -0.0038 | -0.0008 | -0.0027 | -0.0027 | 0.0191 | 0.0058 |
| | L2P | 7.2695 | 3.5268 | 4.6789 | 0.0030 | 0.0054 | 0.0094 | 0.0082 | 0.0342 | 0.0096 |
| | TST | 0.3456 | 0.3051 | 0.2469 | -0.0097 | -0.0026 | 0.0075 | 0.0078 | 0.0133 | -0.0062 |

Table 2: Forgetting rate - averages over 5 random sequences. Positive forgetting rate indicates forgetting while negative forgetting rate indicates knowledge transfer.

(we can see it is even poorer than ONE (prompt), indicating that its selection causes forgetting). CAT performs better, but still has a large gap compared to TST, due to its inaccurate similarity detection and ineffectiveness in handling previous tasks. CTR is the best one among the three, but it is still worse than TST because of the inaccurate instance-level similarity detection.

(4). Sub-network masking on adapters helps TST achieve good results. TST uses adapters and we can see ONE (Adapter) is much better than ONE (Prompt) and is similar to ONE. The performance of prompt-based CL baseline L2P is much worse than other baselines. This is because prompts do not have sufficient trainable parameters, which are also randomly initialized and can be difficult to train.

**Knowledge transfer and forgetting prevention.** To validate TST's effectiveness in dealing with forgetting with a sequence of dissimilar tasks, we compute the *Forgetting Rate* (Liu et al., 2020), $\text{FR} = \frac{1}{t-1} \sum_{i=1}^{t-1} A_{i,i} - A_{t,i}$[13], where $A_{i,i}$ is the forward performance of task $i$ and $A_{t,i}$ is the performance of task $i$ after training the last task $t$. We average over all tasks except the last one because the last task obviously has no forgetting. We report the forward results (TST (forward)) and forgetting rate FR (averaged over 5 random task sequences) for all datasets in Table 2. Regarding forward performance, TST (forward) is clearly better than ONE, indicating effective forward transfer (learned tasks help the new task). Regarding backward transfer, We can see negative forgetting rates in ASC and NER, indicating some positive backward transfer (the training of a new task helps some old tasks). For the other 3 dissimilar datsets, we can see that TST has only slight forgetting. We can also see that the baselines have much larger forgetting comparing to TST. One exception is SupSup, which has no forgetting because it trains different masks/sub-networks for different tasks. This is certainly good for forgetting prevention but makes knowledge transfer impossible.

**Effectiveness of task similarity detection.** Comparing the forgetting rate in Table 2, we can see that TST has only a very slight forgetting compared to baselines, indicating the effectiveness of its

---

[13]Mehta et al. (2021) defined a different forgetting rate. Appendix G will argue that ours is more effective.

| Scenario | Data Model | SUM | | | ASC | | CCD | | DRG | NER |
| | | R1 | R2 | RL | MF1 | Acc | MF1 | Acc | BLEU | F1 |
|---|---|---|---|---|---|---|---|---|---|---|
| Non-CL | ONE | 39.0738 | 10.7076 | 35.2501 | 0.8555 | 0.9150 | 0.9107 | 0.9109 | 0.2414 | 0.5933 |
| | ONE (Adapter) | 38.9000 | 11.5449 | 35.2323 | 0.8395 | 0.9090 | 0.9078 | 0.9081 | 0.2342 | 0.5669 |
| | ONE (Prompt) | 30.6709 | 7.2285 | 27.5339 | 0.7646 | 0.8554 | 0.8623 | 0.8628 | 0.1267 | 0.4590 |
| CL | TST (w/o pool) | 29.9746 | 7.1159 | 27.2446 | 0.9088 | 0.9427 | 0.8760 | 0.8778 | 0.1959 | 0.4836 |
| | TST (w/o similarity) | 38.3660 | 11.6289 | 34.7497 | 0.8898 | 0.9335 | 0.9098 | 0.9100 | **0.2471** | 0.5893 |
| | TST (w/o comparison) | 34.8660 | 9.6079 | 31.4729 | 0.9084 | 0.9421 | 0.8931 | 0.8923 | 0.2180 | 0.5493 |
| | TST | **38.7758** | **11.3652** | **34.9935** | **0.9161** | **0.9453** | **0.9104** | **0.9103** | 0.2425 | **0.6169** |

Table 3: Ablation experiment results - averages over 5 random sequences (the standard deviation is reported in Appendix E). We bold the best performance within CL baselines.

| Scenario | Data Model | SUM | | | ASC | | CCD | | DRG | NER |
| | | R1 | R2 | RL | MF1 | Acc | MF1 | Acc | BLEU | F1 |
|---|---|---|---|---|---|---|---|---|---|---|
| CL | TST (w/o pool) | 8.9037 | 4.7496 | 5.7680 | -0.0055 | -0.0008 | 0.0413 | 0.0400 | 0.0494 | 0.1603 |
| | TST (w/o similarity) | 0.0000 | 0.0000 | 0.0000 | 0.0000 | 0.0000 | 0.0000 | 0.0000 | 0.0000 | 0.0000 |
| | TST (w/o comparison) | 2.7010 | 1.4196 | 1.9688 | -0.0079 | -0.0039 | 0.0202 | 0.0214 | 0.0299 | 0.0850 |
| | TST | 0.3456 | 0.3051 | 0.2469 | -0.0097 | -0.0026 | 0.0075 | 0.0078 | 0.0133 | -0.0062 |

Table 4: Forgetting rate for ablation experiment results.

similarity detection. CAT, CTR and L2P all have much more forgetting than TST except on the CCD data, indicating they have poorer similarity detection. To provide more evidences, we show the task similarity detection results for TST in Appendix H. For *NER*, we see that tasks "conll2003", "wikigold" and "btc" use the same mask (indicating they are similar). This makes sense as they all share the same named entity classes while the other tasks have different named entities. In contrast, CAT, which is the only baseline that does task-level similarity detection, finds all NER tasks dissimilar. In *SUM*, TST finds "icsi" and "ami" to be similar (sharing a mask). This is also reasonable because both tasks are summarization on meetings while the others are conversations of diverse forms. Similarly, TST finds "Yelp" and "Amazon" to be similar in *CCD* because they are all sentiment classification tasks. CAT finds no similarity in both CCD and SUM tasks and produces worse results. Regarding *ASC* and *DRG*, it is hard to say which system is better as ASC tasks are all somehow similar and all tasks of DRG are dissimilar, but we see from the results in Table 1 that TST is better. Similarity detection results of different orders and random seeds are reported in Appendix I.

**Ablation study.** We want to know whether (1) the proposed mask pool, (2) the similarity detection and (3) the importance comparison are helpful. To answer (1), we conduct the ablation experiment **TST (w/o pool)**, where we use only one single mask/sub-network in the pool for all tasks. To answer (2), we conduct the experiment **TST (w/o similarity)**, where we remove the similarity detection and use different masks for different tasks (thus no CF or KT). To answer (3), we conduct experiment **TST (w/o comparison)**, where we remove the importance comparison and simply use the mask with the highest importance based on the current task data as the mask for learning the current task.

Tables 3 and 4 show the ablation results and the corresponding forgetting rates. We can see that the full TST gives the best average result, indicating that every component helps. Additional observations are: (1) TST's gain is partially from the mask pool as TST (w/o pool) is poorer on average, particularly for those datasets having little shared knowledge; (2) Similarity detection helps as TST (w/o similarity) gives worse performance. (3) Our importance comparison is effective. We see that TST (w/o comparison) is poorer, indicating the importance comparison learning in TST helps mitigate CF.

## 5 CONCLUSION

This paper studied task continual learning (Task-CL) on a wide range of NLP problems using pre-trained BART model as backbone to achieve both CF prevention and knowledge transfer (KT). The key novelty of the proposed technique TST is the novel task similarity detection method based on mask importance. The importance is computed to find a similar task so that effective KT and CF prevention can be achieved. CF is prevented by using a different mask while KT is achieved by sharing the same mask. Experimental results showed that TST markedly improves the performance of both the new task and the old tasks via KT and is also effective at overcoming CF.

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

## A  ADDITIONAL DETAILS ABOUT ADAPTER

TST leverages adapter to do masking. An adapter is trainable parameters inserted to each Transformer layers, which adapts the output distribution of a pre-trained LM *without* modifying its original weights (the original LM is fixed). An adapter block is simply a 2-layer fully connected network with layer normalization and residual connections. Figure 2 illustrate the LM with adapter.

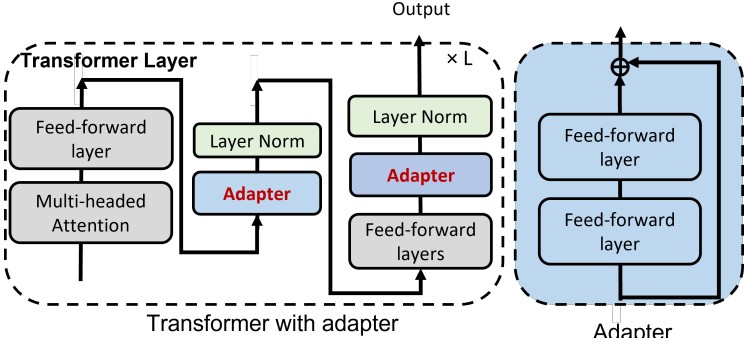

Figure 2: Architecture of Transformer with adapters. An adapter (blue component) is inserted in each layer. Only the blue and green boxes are trainable while the LM is fixed during training.

## B  ADDITIONAL DETAILS ABOUT MASK TRAINING

In Sec. 3.1, we mentioned that we train a mask to identify a sub-network for a task without training the adapter's weights/parameters ($W$). We follow the idea in Wortsman et al. (2020). Specifically, to train the mask, we learn a randomly initialized score $s_{i,j}^{(k)}$ for each entry/parameter $w_{i,j}$ in $W$. Once trained, these scores are *thresholded* to obtain the mask that can indicate a sub-network. Therefore, the $M^{(k)}$ can be seen as,

$$M_{i,j}^{(k)} = h(s_{i,j}^{(k)}) \tag{8}$$

where $h$ is a function which outputs 1 for top-$n\%$ of the scores in the layer with $n$ as a pre-defined mask density (we use 80 following the Wortsman et al. (2020)). To make the scores differentiable, we leverage the "straight-through" trick (Bengio et al., 2013) to update the scores as follows,

$$s_{i,j}^{(k)} = s_{i,j}^{(k)} - \alpha \nabla_{s_{i,j}^{(k)}}$$
$$\nabla_{s_{i,j}^{(k)}} = \frac{\partial \mathcal{L}}{\partial s_{i,j}^{(k)}} = \frac{\partial \mathcal{L}}{\partial \mathcal{I}_j} \frac{\partial \mathcal{I}_j}{\partial s_{i,j}^{(k)}} = \frac{\partial \mathcal{L}}{\partial \mathcal{I}_j} w_{i,j} \mathcal{O}_i \tag{9}$$

where $\mathcal{I}_j$ and $\mathcal{O}_i$ refer to the input and output of the units $i$ and $j$ respectively. We note this is efficient because the adapter size is small. On the disk, we need to store the additional Boolean mask which takes only 1-bit for each parameter. During continual training, it is possible that we select a trained mask (Sec. 3.3). In this case, the stored binary mask can be used to initialize the score $s_{i,j}^{(k)}$.

## C  ADDITIONAL DETAILS ABOUT THE DATASETS

Recall that TST uses five datasets. Here we give their detailed statistics.

(1) **ASC.** ASC is more than a traditional classification problem because of the additional input of the aspect and the fact that in the same sentence different aspects can have different opinions. Here we show more detailed on these datasets in Table 5.

(2) **CCD, SUM, DRG and NER.** We give their detailed statistic in Table 6

| Tasks/Domains | #Training | #Validating | #Testing |
|---|---|---|---|
| Speaker | 233 S./352 A./287 P./65 N./0 Ne. | 30 S./44 A./35 P./9 N./0 Ne. | 38 S./44 A./40 P./4 N./0 Ne. |
| Router | 200 S./245 A./142 P./103 N./0 Ne. | 24 S./31 A./19 P./12 N./0 Ne. | 22 S./31 A./24 P./7 N./0 Ne. |
| Computer | 187 S./283 A./218 P./65 N./0 Ne. | 25 S./35 A./23 P./12 N./0 Ne. | 29 S./36 A./29 P./7 N./0 Ne. |
| Nokia6610 | 209 S./271 A./198 P./73 N./0 Ne. | 29 S./34 A./30 P./4 N./0 Ne. | 28 S./34 A./25 P./9 N./0 Ne. |
| Nikon4300 | 131 S./162 A./135 P./27 N./0 Ne. | 15 S./20 A./18 P./2 N./0 Ne. | 15 S./21 A./19 P./2 N./0 Ne. |
| Creative | 582 S./677 A./422 P./255 N./0 Ne. | 68 S./85 A./42 P./43 N./0 Ne. | 70 S./85 A./52 P./33 N./0 Ne. |
| CanonG3 | 190 S./228 A./180 P./48 N./0 Ne. | 25 S./29 A./21 P./8 N./0 Ne. | 24 S./29 A./24 P./5 N./0 Ne. |
| ApexAD | 281 S./343 A./146 P./197 N./0 Ne. | 35 S./43 A./16 P./27 N./0 Ne. | 28 S./43 A./31 P./12 N./0 Ne. |
| CanonD500 | 103 S./118 A./96 P./22 N./0 Ne. | 11 S./15 A./14 P./1 N./0 Ne. | 13 S./15 A./11 P./4 N./0 Ne. |
| Canon100 | 137 S./175 A./123 P./52 N./0 Ne. | 19 S./22 A./20 P./2 N./0 Ne. | 16 S./22 A./21 P./1 N./0 Ne. |
| Diaper | 166 S./191 A./143 P./48 N./0 Ne. | 22 S./24 A./18 P./6 N./0 Ne. | 24 S./24 A./22 P./2 N./0 Ne. |
| Hitachi | 152 S./212 A./153 P./59 N./0 Ne. | 23 S./26 A./19 P./7 N./0 Ne. | 23 S./27 A./14 P./13 N./0 Ne. |
| Ipod | 124 S./153 A./101 P./52 N./0 Ne. | 18 S./19 A./14 P./5 N./0 Ne. | 19 S./20 A./15 P./5 N./0 Ne. |
| Linksys | 152 S./176 A./128 P./48 N./0 Ne. | 19 S./22 A./13 P./9 N./0 Ne. | 20 S./23 A./16 P./7 N./0 Ne. |
| MicroMP3 | 384 S./484 A./340 P./144 N./0 Ne. | 42 S./61 A./48 P./13 N./0 Ne. | 51 S./61 A./39 P./22 N./0 Ne. |
| Nokia6600 | 298 S./362 A./244 P./118 N./0 Ne. | 26 S./45 A./32 P./13 N./0 Ne. | 39 S./46 A./30 P./16 N./0 Ne. |
| Norton | 168 S./194 A./54 P./140 N./0 Ne. | 17 S./24 A./15 P./9 N./0 Ne. | 24 S./25 A./5 P./20 N./0 Ne. |
| Restaurant | 1893 S./3452 A./2094 P./779 N./579 Ne. | 84 S./150 A./70 P./26 N./54 Ne. | 600 S./1120 A./728 P./196 N./196 Ne. |
| Laptop | 1360 S./2163 A./930 P./800 N./433 Ne. | 98 S./150 A./57 P./66 N./27 Ne. | 411 S./638 A./341 P./128 N./169 Ne. |

Table 5: Statistics of the ASC tasks. **S.**: number of sentences; **A**: number of aspects; **P., N., and Ne.**: number aspects with positive, negative and neutral opinions, respectively. Note that the "Restaurant" and "Laptop" have 3 classes of opinion polarities (positive, negative and neutral) while the others have only 2 classes (positive and negative).

| Dataset | Tasks/Domains | #Training | #Validating | #Testing | #Classes |
|---|---|---|---|---|---|
| CCD | Yahoo | 4500 | 500 | 4840 | 10 |
| | AGnews | 1785 | 199 | 1756 | 2 |
| | Amazon | 898 | 100 | 998 | 2 |
| | Dbpedia | 6237 | 693 | 6748 | 14 |
| | Yelp | 900 | 100 | 984 | 4 |
| SUM | icsi | 43 | 10 | 6 | — |
| | ami | 97 | 20 | 20 | — |
| | reddit | 201 | 50 | 250 | — |
| | stack | 205 | 50 | 250 | — |
| | nyt | 200 | 50 | 250 | — |
| | emails | 215 | 50 | 250 | — |
| DRG | taxi | 406 | 71 | 56 | — |
| | hotel | 3366 | 143 | 177 | — |
| | attraction | 298 | 27 | 28 | — |
| | train | 1954 | 196 | 262 | — |
| | restaurant | 569 | 63 | 59 | — |
| NER | conll2003 | 200 | 3250 | 3453 | 9 |
| | wikigold | 200 | 170 | 170 | 9 |
| | btc | 200 | 934 | 934 | 9 |
| | re3d | 200 | 77 | 200 | 21 |
| | gum | 200 | 250 | 1000 | 23 |

Table 6: Statistics of the CCD, SUM, DRE and NER datasets. Number of classes is not applicable to SUM and DRG because they are generation datasets.

# D HYPERPARAMETERS

Unless otherwise stated, the same hyper-parameters are used in all experiments. The maximum input length is set to 128 for all datasets except for SUM which uses 1024 due to its longer sequences. `AdamW` optimizer is used. The learning rate is set to 5e-5 for Transformer (search within {5e-2,5e-3,5e-4,5e-5}), 3e-2 for prompt (search within {3e-1,3e-2,3e-3,3e-4,3e-5}), adapter and classifier. The prompt length is set to 20 (search within {10,20,50,80,100,150}) and adapter bottleneck size is 64, following the original papers (Houlsby et al., 2019). The batch size is set to 32 and the number of training epochs is set to 50 with early stopping. The number of mask $K$ in the mask pool is set to the same as the number of task. For classification tasks, a separate classification head is used for each task in the sequence. For generation tasks, a shared LM head is used for all tasks in a sequence. we further set the number of beams to 4 for beam search and constrain the target length in between 30 to 200. For image-based (EWC, HAT, SupSup, CAT, L2P) and RoBERTa-based (CTR) systems, we

adapt them for text classification and generation by replacing their feature extractors with BART. For LAMOL, we directly run the author provided code. Except for the aforementioned hyper-parameters, all baseline-specific hyper-parameters follow those in their original papers.

## E  STANDARD DEVIATIONS

Table 7 reports the standard deviations of the corresponding results in Table 1 (in the main paper) of TST and the considered baselines over 5 runs with random sequences. We only report the CL baselines since they are related to the task order. We can see the results of TST is stable. SupSup trains different mask/sub-networks for different tasks, so it is not related to the task order and not reported in the table.

Table 8 reports the standard deviations of the corresponding results in Table 3 (in the main paper) of TST and the considered baselines over 5 runs with random sequences. We can see the results of TST and its variants are stable. TST (w/o similarity) trains different mask/sub-network for different tasks, so it is not related to task order and not reported in the table.

| Scenario | Data Model | SUM | | | ASC | | CCD | | DRG | NER |
|---|---|---|---|---|---|---|---|---|---|---|
| | | R1 | R2 | RL | MF1 | Acc | MF1 | Acc | BLEU | F1 |
| CL | NCL | ±0.6028 | ±0.5458 | ±0.6623 | ±0.0040 | ±0.0027 | ±0.0233 | ±0.0230 | ±0.0060 | ±0.0767 |
| | EWC | ±1.3205 | ±0.5331 | ±1.1258 | ±0.0093 | ±0.0063 | ±0.0227 | ±0.0216 | ±0.0217 | ±0.0707 |
| | HAT | ±1.3035 | ±0.4912 | ±1.1464 | ±0.0130 | ±0.0042 | ±0.0028 | ±0.0028 | ±0.0094 | ±0.0063 |
| | LAMOL | ±1.2519 | ±0.4218 | ±0.5231 | ±0.0085 | ±0.0039 | ±0.0317 | ±0.0258 | ±0.0068 | — |
| | CAT | ±1.0424 | ±0.4027 | ±0.9029 | ±0.0096 | ±0.0021 | ±0.0019 | ±0.0021 | ±0.0051 | ±0.0106 |
| | CTR | ±1.1595 | ±0.5189 | ±0.9646 | ±0.0080 | ±0.0030 | ±0.0009 | ±0.0010 | ±0.0127 | ±0.0049 |
| | L2P | ±2.1496 | ±0.4747 | ±2.0267 | ±0.0650 | ±0.0361 | ±0.0181 | ±0.0176 | ±0.0216 | ±0.0022 |
| | TST | ±0.5287 | ±0.3585 | ±0.5175 | ±0.0027 | ±0.0032 | ±0.0049 | ±0.0048 | ±0.0079 | ±0.0031 |

Table 7: Standard deviations of the corresponding metrics of the proposed TST model and the baselines on the five different datasets.

| Scenario | Data Model | SUM | | | ASC | | CCD | | DRG | NER |
|---|---|---|---|---|---|---|---|---|---|---|
| | | R1 | R2 | RL | MF1 | Acc | MF1 | Acc | BLEU | F1 |
| CL | TST (w/o pool) | ±0.8603 | ±0.3504 | ±0.5482 | ±0.0049 | ±0.0034 | ±0.0300 | ±0.0287 | ±0.0103 | ±0.0437 |
| | TST (w/o comparison) | ±1.0655 | ±0.5163 | ±0.9594 | ±0.0073 | ±0.0028 | ±0.0098 | ±0.0089 | ±0.0057 | ±0.0374 |
| | TST | ±0.5287 | ±0.3585 | ±0.5175 | ±0.0027 | ±0.0032 | ±0.0049 | ±0.0048 | ±0.0079 | ±0.0031 |

Table 8: Standard deviations of the corresponding metrics of the proposed TST model and the six ablation experiments.

## F  RESULTS FOR INDIVIDUAL SEQUENCES

In Table 1, we report the results averaged over 5 random sequences (different task orders). In this section, we give the results of each sequence in Table 9. We can see that the order indeed affects the results but not by much. In summary, we believe the average over random sequences in Table 1 can show us the effectiveness of TST.

| Order | SUM | | | ASC | | CCD | | DRG | NER |
|---|---|---|---|---|---|---|---|---|---|
| | R1 | R2 | RL | MF1 | Acc | MF1 | Acc | BLEU | F1 |
| 1 | 39.0532 | 11.7343 | 35.4234 | 0.9144 | 0.9497 | 0.9102 | 0.9104 | 0.2422 | 0.6137 |
| 2 | 38.4992 | 11.0726 | 34.7320 | 0.9171 | 0.9435 | 0.9081 | 0.9084 | 0.2429 | 0.6137 |
| 3 | 37.9608 | 10.7857 | 34.0924 | 0.9200 | 0.9466 | 0.9036 | 0.9038 | 0.2376 | 0.6216 |
| 4 | 37.6063 | 10.7691 | 34.0197 | 0.9130 | 0.9412 | 0.9100 | 0.9093 | 0.2472 | 0.6163 |
| 5 | 38.7750 | 11.2888 | 34.8252 | 0.9005 | 0.9454 | 0.9109 | 0.9112 | 0.2396 | 0.6192 |

Table 9: Results for individual domains.

## G  DIFFERENCE BETWEEN OUR FORGETTING RATE AND THE ONE IN (MEHTA ET AL., 2021)

Unlike our forgetting rate in Sec. 4, the forgetting rate in (Mehta et al., 2021) is defined as $F'_t = \frac{1}{t-1} \sum_{\tau=1}^{t-1} \max_{\tau' \in \{1,...,t-1\}} (S_{\tau',\tau} - S_{t,\tau})$. Since both metrics measure everything from the standing

point of the end of continual learning, i.e., after all tasks are learned, we believe our measure is more reasonable. Let us use some examples to illustrate,

- (1). if task 1 archives the accuracy 0.5 right after its training, it achieves 0.4 after task 2, and it achieves 0.3 after task 3 (final task). In this case, both measures give the same forgetting 0.2.

- (2). if task 1 archives the accuracy 0.5 right after its training, it achieves 0.8 after task 2, and it achieves 0.82 after task 3 (final task). If we take max ($F_t'$), then the forgetting is -0.02, but our method will give -0.32. In this case, our method is more reasonable because it precisely shows how much backward transfer (negative value here means backward transfer) has been achieved for task 1 after task 2 and task 3 are learned since both measures evaluate from the same reference point, i.e., after all tasks are learned (the last task is t in both measures, i.e., task 3 in this example).

- (3). If task 1 archives the accuracy 0.5 right after its training, 0.8 after task 2 and 0.4 after task 3 (final task), our metric will give the forgetting 0.1. $F_t'$ will give 0.4. This case is more debatable because $F_t'$ catches the worst forgetting. But since we evaluate after all tasks are learned (the reference point is when the last task is learned), again we believe that our method is more reasonable.

- (4). If task 1 archives the accuracy 0.5 right after its training, 0.1 after task 2 and 0.4 after task 3 (final task), both metrics will give the forgetting 0.1. In this case, $F_t'$ does not catch the worst forgetting of 0.4 (0.5-0.1) in the process. Again, if we agree that we evaluate from the reference point of when the last task is learned, then both measures are fine in this case.

In summary, while we believe ours is more reasonable, a better metric may be designed in the future to characterize forgetting and knowledge transfer in the continual learning process.

## H  COMPARING TASK SIMILARITY DETECTION RESULTS BETWEEN CAT AND TST

In Sec. 4, we discussed the effectiveness of task similarity comparison and compare the similarity detection results of TST and CAT. Figure 4 show the task similarity results for TST. Figure 3 shows the task similarity results on the ASC data for CAT. CAT did not find any similar tasks for the other datasets. The analysis is already in Sec. 4.

## I  DETAILED SIMILARITY DETECTION RESULTS FOR TST IN DIFFERENT SEQUENCES AND RANDOM SEEDS

In this section, we are interested in how the random seeds and the task orders affect the detected similarity.

We give the detected similar and dissimilar tasks for all datasets in Table 11. The detailed sequences for these datasets are given in Table 10. We can observe the followings:

(1) The similarity detection results in different task orders and different seeds have some differences but not much. It is reasonable to have differences as the learning of tasks is quite dynamic.

(2) The effect of random seeds and orders is different across datasets. For example, in SUM, the similarity detection results in different seeds have differences, which is probably because each task in SUM is a long document and has more information and easier to detect the similarity. In other datasets, We can see different orders and seeds made some differences, indicating that it is more difficult to detect similarity for them, which is probably because the tasks in these datasets are sentence-based text classification (extraction or generation) and the information contained in a sentence is limited. Thus, the detection has a larger variance.

In summary, as we discussed in Sec. 4), the task similarity is fuzzy and it is hard to say which task similarity is actually correct or wrong as there are no ground-truth similarity labels. However, the average results over random sequences in Table 1 show us the effectiveness of TST.

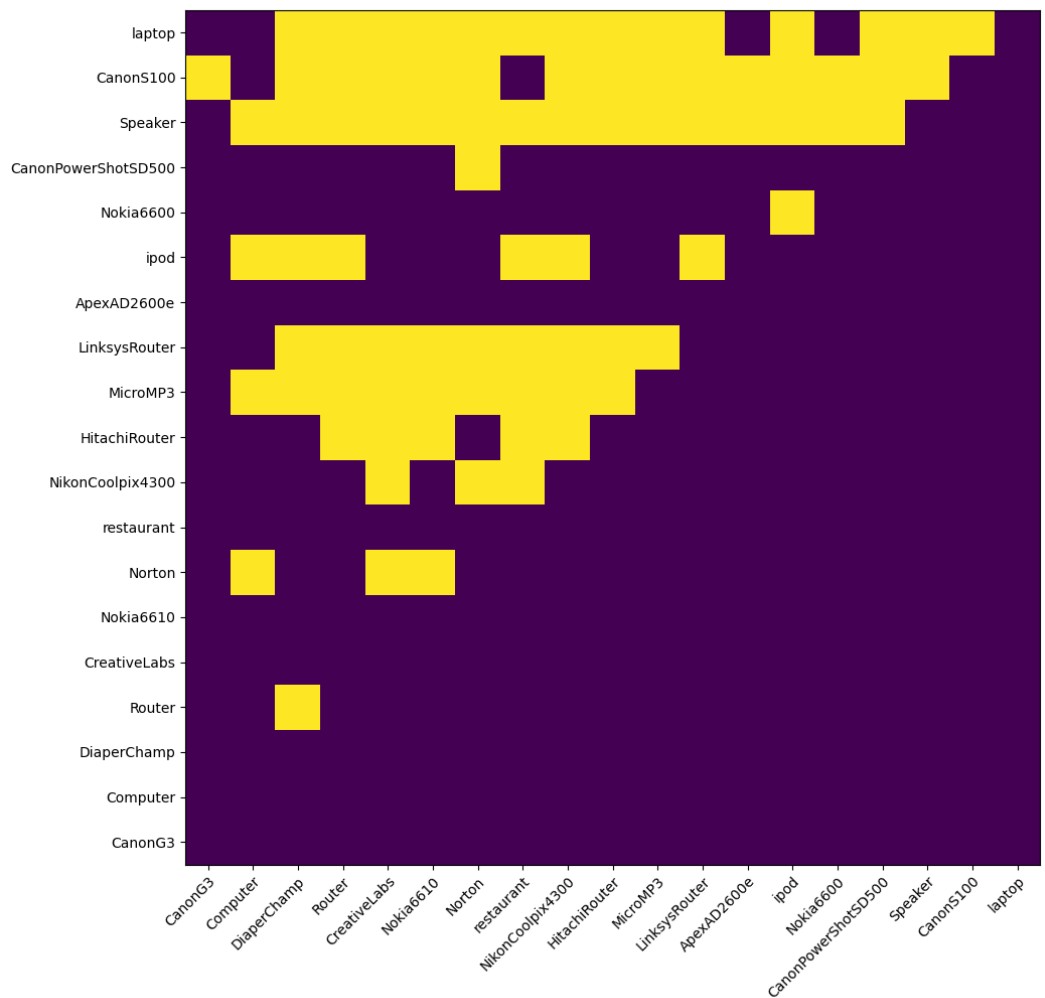

Figure 3: Similarity detection results for ASC of CAT. The y-axis gives the sequence of tasks that are learned, and the x-axis lists the same set of task names. The yellow cells indicate that the corresponding tasks are detected by CAT as similar (one task can be similar to multiple tasks but only the previously learned tasks are considered for the current task) while the purple cells indicate the tasks are dissimilar or unknown (because they have not been learned). For example row 3 means when we are learning task "DiaperChamp", CAT finds it to be dissimilar to any of learned tasks ("CanonG3" and "Computer"); Row 4 means when we are learning task "Router", CAT finds it to be similar to task "DiaperChamp".

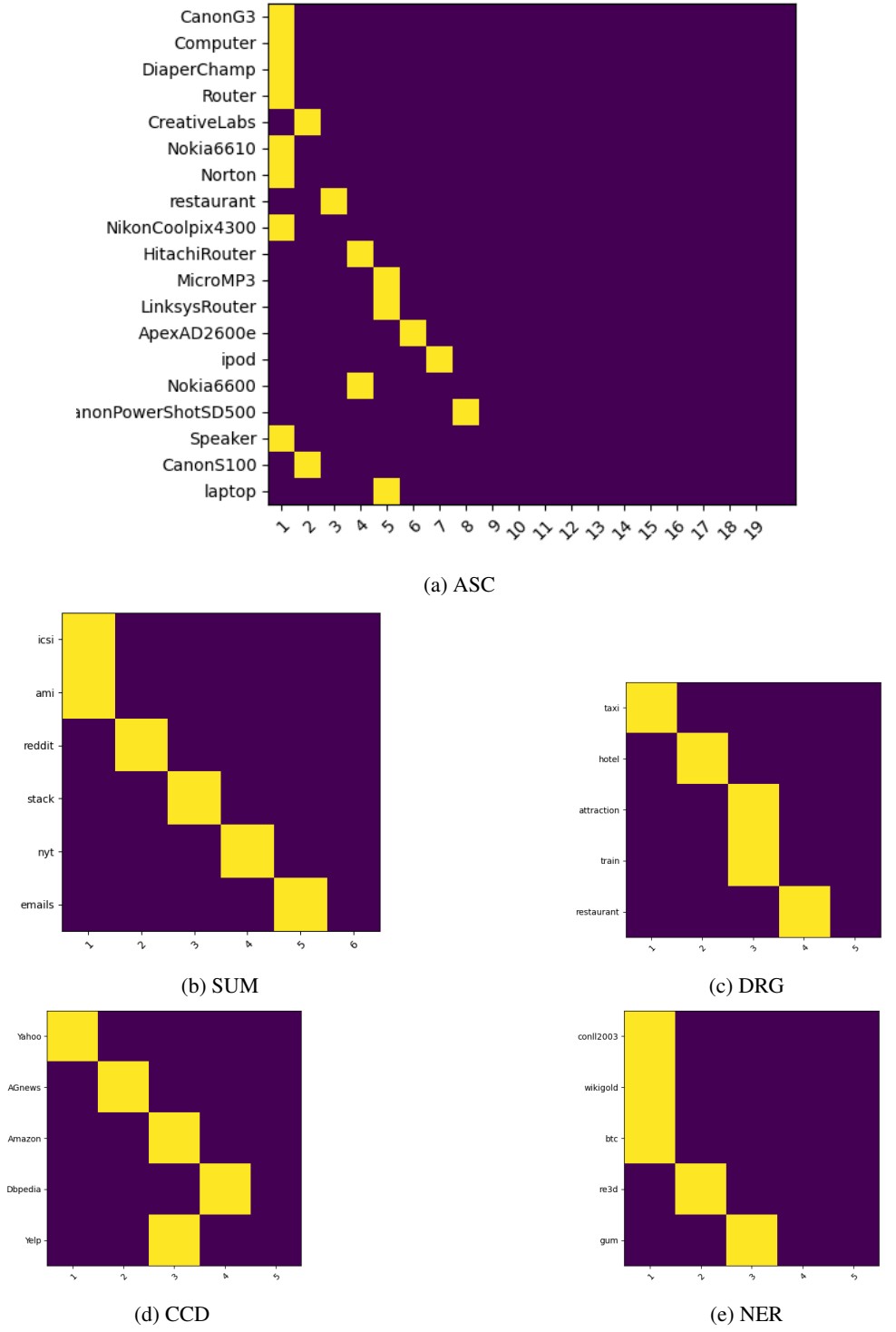

Figure 4: Similarity detection results of TST (after the continual training of all tasks). X-axis shows mask IDs and y-axis shows the task names. The yellow cells indicate that the masks are used by some tasks while the purple cells indicate the masks are not used by their corresponding tasks. If multiple tasks use the same mask, they are regarded as similar by our system TST. Note that not all masks are used because different tasks may share the same mask.

| Dataset | Order | Sequences |
|---------|-------|-----------|
| SUM | 1 | icsi → ami → reddit → stack → nyt → emails |
|  | 2 | nyt → stack → reddit → emails → ami → icsi |
|  | 3 | emails → ami → reddit → nyt → stack → icsi |
|  | 4 | stack → nyt → reddit → emails → ami → icsi |
|  | 5 | reddit → icsi → nyt → ami → stack → emails |
| ASC | 1 | CanonG3 → Computer → DiaperChamp → Router → CreativeLabs → Nokia6610 → Norton → rest → Nikon4300 → HitachiRouter → MicroMP3 → LinksysRouter → ApexAD2600 → ipod → Nokia6600 → CanonD500 → Speaker → CanonS100 → laptop |
|  | 2 | laptop → rest → HitachiRouter → CanonS100 → ipod → ApexAD2600 → Nokia6600 → CanonD500 → CreativeLabs → Norton → MicroMP3 → Speaker → LinksysRouter → Nokia6610 → Nikon4300 → CanonG3 → Computer → Router → DiaperChamp |
|  | 3 | CanonD500 → LinksysRouter → Nikon4300 → Norton → Computer → MicroMP3 → ApexAD2600 → rest → ipod → CanonG3 → laptop → Nokia6610 → HitachiRouter → Speaker → DiaperChamp → CanonS100 → Router → Nokia6600 → CreativeLabs |
|  | 4 | CanonG3 → Computer → CanonD500 → Nokia6600 → Nikon4300 → LinksysRouter → ApexAD2600 → Router → Speaker → laptop → CanonS100 → rest → Norton → CreativeLabs → ipod → Nokia6610 → MicroMP3 → DiaperChamp → HitachiRouter |
| CCD | 1 | yahoo → agnews → amazon → dbpedia → yelp |
|  | 2 | yahoo → yelp → amazon → dbpedia → agnews |
|  | 3 | amazon → yelp → yahoo → agnews → dbpedia |
|  | 4 | yahoo → dbpedia → yelp → agnews → amazon |
|  | 5 | yahoo → yelp → dbpedia → amazon → agnews |
| DRG | 1 | taxi → hotel → attraction → train → restaurant |
|  | 2 | train → attraction → taxi → hotel → restaurant |
|  | 3 | attraction → hotel → restaurant → taxi → train |
|  | 4 | taxi → attraction → hotel → restaurant → train |
|  | 5 | restaurant → attraction → train → taxi → hotel |
| NER | 1 | conll2003 → wikigold → btc → re3d → gum |
|  | 2 | btc → conll2003 → wikigold → re3d → gum |
|  | 3 | wikigold → re3d → conll2003 → gum → btc |
|  | 4 | btc → conll2003 → wikigold → gum → re3d |
|  | 5 | wikigold → re3d → gum → conll2003 → btc |

Table 10: Random sequence orders of the 5 datasets.

| Dataset | | | Similar tasks | Disismilar tasks |
|---|---|---|---|---|
| SUM | Order | 1 | {ami, icsi} | nyt, stack,reddits, emails |
| | | 2 | {ami, icsi} | nyt, stack,reddits, emails |
| | | 3 | {ami, icsi} | nyt, stack,reddits, emails |
| | | 4 | {ami, icsi} | nyt, stack,reddits, emails |
| | | 5 | — | ami, icsi, nyt, stack,reddits, email |
| | Seed | 1 | {ami, icsi} | nyt, stack,reddits, emails |
| | | 2 | {ami, icsi} | nyt, stack,reddits, emails |
| | | 3 | {ami, icsi} | nyt, stack,reddits, emails |
| | | 4 | {ami, icsi} | nyt, stack,reddits, emails |
| | | 5 | {ami, icsi} | nyt, stack,reddits, emails |
| ASC | Order | 1 | {CanonG3, Computer, DiaperChamp, Router, Nokia6610, Norton, Nikon4300, Speaker}, {Creativelab, CanonS100} {MicroMP3, LinksysRouter, laptop}, {HitachiRouter, Nokia6600} | ApexAD2600, rest, ipod, CanonD500 |
| | | 2 | {HitachiRouter, CanonS100, CanonD500, CreativeLabs, Speaker,Nokia6610, DiaperChamp }, {ipod, Nokia6600, Norton, LinksysRouter, Nikon4300, Computer}, {Router, CanonG3} | laptop, rest, ApexAD2600, MicroMP3, |
| | | 3 | {LinksysRouter, Nikon4300, Norton, ApexAD2600}, {MicroMP3, HitachiRouter, DiaperChamp, CanonS100}, {rest, ipod, Nokia6610, Speaker}, {CanonG3, laptop} | CanonD500, CreativeLabs, Nokia6600, Router, Computer |
| | | 4 | {Computer, CanonD500, Nokia6600, Nikon4300, CreativeLabs, ipod, Nokia6610, HitachiRouter}, {LinksysRouter,}, {laptop, CanonS100, MicroMP3}, {rest, DiaperChamp}, {Norton, ApexAD2600} | CanonG3, Router, Speaker |
| | | 5 | {Router, Computer, rest}, {LinksysRouter, DiaperChamp, CanonD500}, {laptop, Nokia6600, HitachiRouter, CanonS100, Nokia6610}, {Nikon4300, ipod, CreativeLabs, CanonG3} | Norton, ApexAD2600, MicroMP3, Speaker |
| | Seed | 1 | {HitachiRouter, CanonS100, CanonD500, CreativeLabs, Speaker,Nokia6610, DiaperChamp}, {ipod, Nokia6600, Norton, LinksysRouter, Nikon4300, Computer}, {Router, CanonG3} | laptop, rest, ApexAD2600, MicroMP3, |
| | | 2 | {rest, HitachiRouter, ipod, Nikon4300, CanonG3, MicroMP3}, {CanonS100, CanonD500, Speaker, Computer, DiaperChamp}, {Norton, Nokia6610} | laptop, Router, LinksysRouter, ApexAD2600, Nokia6600, CreativeLabs |
| | | 3 | {rest, ipod, ApexAD2600}, {HitachiRouter, CanonS100, Nokia6600}, {CanonD500, Nokia6610, DiaperChamp}, {CreativeLabs, Norton}, {MicroMP3, LinksysRouter, CanonG3}, {Nikon4300, Computer} | laptop, Router, Speaker |
| | | 4 | {ipod, ApexAD2600}, {CanonD500, DiaperChamp}, {CreativeLabs, MicroMP3, Speaker, LinksysRouter}, {CanonG3, Router} | laptop, rest, HitachiRouter, CanonS100, Nokia6600, Norton, Nokia6610, Nikon4300, Computer |
| | | 5 | {rest, HitachiRouter, CanonS100, ipod, Norton}, {ApexAD2600, CanonD500, Speaker}, {MicroMP3, LinksysRouter}, {Nikon4300, CanonG3}, {Nokia6610, Router} | laptop, Nokia6600, CreativeLabs, Computer, DiaperChamp |
| CCD | Order | 1 | {yelp, amazon} | yahoo, dbpedia, agnews |
| | | 2 | {yelp, amazon} | yahoo, dbpedia, agnews |
| | | 3 | {yelp, amazon}, {yahoo, agnews} | dbpedia |
| | | 4 | {yahoo, dbpedia, yelp} | agnews, amazon |
| | | 5 | {yelp, amazon} | yahoo, dbpedia, agnews |
| | Seed | 1 | {yelp, amazon} | yahoo, dbpedia, agnews |
| | | 2 | {yelp, dbpedia}, {amazon, agnews} | yahoo |
| | | 3 | {yelp, dbpedia} | yahoo, amazon, agnews |
| | | 4 | — | yelp, dbpedia, yahoo, amazon, agnews |
| | | 5 | — | yelp, dbpedia, yahoo, amazon, agnews |
| DRG | Order | 1 | {attraction, train} | taxi, hotel, restaurant |
| | | 2 | {attraction, train} | taxi, hotel, restaurant |
| | | 3 | {attraction, hotel, restaurant} | taxi, train |
| | | 4 | {attraction, train} | taxi, hotel, restaurant |
| | | 5 | — | attraction, train, taxi, hotel, restaurant |
| | Seed | 1 | {attraction, train} | taxi, hotel, restaurant |
| | | 2 | — | attraction, train, taxi, hotel, restaurant |
| | | 3 | {attraction, train} | taxi, hotel, restaurant |
| | | 4 | — | attraction, train, taxi, hotel, restaurant |
| | | 5 | — | attraction, train, taxi, hotel, restaurant |
| NER | Order | 1 | {conll2003, wikigold, btc} | re3d, gum |
| | | 2 | — | conll2003, wikigold, btc, re3d, gum |
| | | 3 | {btc, re3d} | conll2003, wikigold, gum |
| | | 4 | — | conll2003, wikigold, btc, re3d, gum |
| | | 5 | — | conll2003, wikigold, btc, re3d, gum |
| | Seed | 1 | {conll2003, wikigold, btc} | re3d, gum |
| | | 2 | — | conll2003, wikigold, btc, re3d, gum |
| | | 3 | {re3d, conll2003} | wikigold, gum, btc |
| | | 4 | {btc, re3d} | conll2003, wikigold, gum |
| | | 5 | — | conll2003, wikigold, btc, re3d, gum |

Table 11: Similarity detection results for all 5 datasets in different orders and different seeds (based on order 1). Tasks in "{}" means they are detected as similar. "–" indicates there is no similar tasks found.

