# OpenReview forum: "Continual Learning Based on Sub-Networks and Task Similarity"
_ICLR.cc/2023/Conference — Submitted to ICLR 2023_

### Official Review · Reviewer_Umdv · 2022-10-24

**Confidence:** 4
**Correctness:** 3
**Technical Novelty And Significance:** 2
**Empirical Novelty And Significance:** 2
**Recommendation:** 5

**Clarity, Quality, Novelty And Reproducibility:**

The technical presentation was hard to follow at times. For example, exactly how masks are applied to parameter sets is not clear from this paper alone; I had to go read the cited papers to understand the ideas being applied here (Wortsman et al, 2020; Houlsby et al., 2019). In several places, the gradient is used to score the importance of different masks, but I had trouble figuring out what the gradient is being evaluated on, and then how it is aggregated across examples to yield a single task importance score. To this end, Figure 1 seeks to illustrate how the whole framework operates, but I didn’t find it particularly helpful (it could also benefit from proper typesetting).

**Strength And Weaknesses:**

The experiments are fairly systematic in terms of considering a large number of baselines and datasets. However, it should be noted that the improvements over certain CL baselines like SupSup (Wortsman et al., 2020) are very similar in performance (in one case, SupSup achieves 11.63, while the proposed method gets 11.36 and is incorrectly labelled in bold as the best performance).

I would have liked to see more motivation for the proposed masking approach. Specifically, looking at the gradients does not strike me as the most obvious or computationally expedient mechanism to compute task similarity. For example, task similarity could be measured instead on the basis of simple statistics of the feature representations (e.g., multivariate Gaussian fit to each task’s activations).  In general, additional discussion of the limitations of the proposed approach and discussion of alternatives would benefit the paper.

I would have liked to see a multi-task “oracle” that has access to all tasks at once during training. This would help characterise the potential performance benefit of knowledge transfer separate from the impact of catastrophic forgetting.


**Summary Of The Paper:**

Prior work in continual learning has focused on addressing the catastrophic forgetting (CF) problem. However, this paper notes that "knowledge transfer" (KT) between tasks is also important, and proposes a new method which "addresses both issues."

To address CF, each task is associated with a learned mask identifying a task-specific sub-network. The mechanism by which this is accomplished is “borrowed from (Wortsman et al., 2020)” and consists of boolean gates indicating which parameters in the network should be employed for which tasks.

To address KT, the idea is to measure the importance of the mask of the previous task relative to the current task. This uses an idea from network pruning (Michel et al., 2019) and uses gradients as measure of parameter importance. A contribution of this work is to extend this idea to entire task-specific masks.

The experiments consist of five benchmarks: (1) 19 tasks for aspect sentiment classification, (2) 5 tasks for continual classification dataset, (3) 6 tasks from ConvoSum, (4) 5 tasks for dialogue response generation and (5) 5 named-entity recognition tasks. On these benchmarks, 11 baselines are compared, including continual learning and non continual learning methods.


**Summary Of The Review:**

Overall, this paper provides a sensible framework for continual learning, providing mechanisms to address both CF and KT. My main concern is the lack of novelty: the main technical contribution is how task similarity is estimated but even this is largely based on prior work. Furthermore, while experiments are fairly comprehensive, in some cases the improvement over baselines is inconsistent or relatively small (e.g., SupSup).

---

> ### Author Response · Authors · 2022-11-18
> **Response to Reviewer Umdv (Part 1)**
>
> > the improvements over certain CL baselines are very similar in performance (SupSup achieves 11.63, while the proposed method gets 11.36 and is incorrectly labelled in bold).
>
> Sorry, we bold the incorrect number. We have corrected it in the revised version.  Regarding the small improvement in some datasets, we answer together with your last question.
>
> > I would have liked to see more motivation for the proposed masking approach. ... looking at the gradients does not strike me as the most obvious or computationally expedient mechanism to compute task similarity..additional discussion of the limitations of the proposed approach and discussion of alternatives would benefit the paper.
>
> This is definitely an interesting question and we have actually thought about it. We have tried many feature-based similarity methods (including those in CAT, CTR and L2P in the baselines) and they work poorly. The key difficulty of these methods (and the method you suggested) is that they all need a threshold to determine what is similar enough to ensure positive knowledge transfer, which is very hard to decide. Further, such a threshold also changes with tasks and datasets. It is very challenging to design an automated way to find a suitable threshold for different scenarios. If a threshold is not selected appropriately, the result will be very bad due to catastrophic forgetting.
>
> The motivation of our mechanism is as follows. As explained (first paragraph in Sec. 3), first of all, the masking approach is able to overcome CF, but it cannot do knowledge transfer. We then proposed a novel method to indirectly compute the task similarity based on the gradient-based importance of a learned mask for a previous task to the current task without using an importance or similarity threshold. Our proposed method works very well. Hope this clears your doubt.
>
> Regarding limitations, there are some that we are planning to work on in our future work. For example, (1) although our current approach works well, the gradient can be sensitive to the data size. It is still not clear whether TST can work well when the amount of data is small. (2) The current knowledge transfer only happens in the same mask shared by two detected similar tasks. Ideally, the knowledge transfer should be more fine-grained and enable partial transfer and transfer among multiple tasks at the same time.
>
> >I would have liked to see a multi-task “oracle” that has access to all tasks at once during training.
>
> Since this is not a continual learning baseline, we did not include it. Regarding knowledge transfer, it is commonly defined as improvement over learning each task separately, i.e., ONE. We have the results in the paper.
>
> As you suggested, we have conducted new experiments to include all tasks at once. For the classification datasets (ASC, CCD and NER), we conducted a multi-task experiment with multi-heads (called “MTL”). For generation datasets (SUM and DRG), it is not possible to do MTL because the language modeling head on top of BART is a linear layer with weights tied to the input embeddings. We thus follow the standard practice (e.g., [a,b]) that pools all data together to train a single shared head model (called “Comb”). The results are added in Table 1. We can see the results of TST are on par with MTL/Comb. This again indicates the effectiveness of TST. In some cases, our system even outperforms MTL/Comb because multitask learning may cause some negative interference among tasks, but in our case, our task similarity mechanism takes care of that.
>
> [a]: Qin et al.,  LFPT5: A Unified Framework for Lifelong Few-shot Language Learning Based on Prompt Tuning of T5, ICLR 2022
>
> [b]: Madotto et al., Continual Learning in Task-Oriented Dialogue Systems, EMNLP 2021

---

> > ### Author Response · Authors · 2022-11-18
> > **Reponse to Reviewer Umdv (Part 2)**
> >
> > > The technical presentation was hard to follow at times
> >
> > We did not give too much detail about the mask training because we follow SupSup (Wortsman et al, 2020) in training and saving the mask (note that SupSup by design is unable to transfer knowledge across tasks). We have made it clearer by adding the training details in Appendix B. We did not introduce the concept of Adapter (Houlsby et al., 2019) as it has been widely used in the NLP area. An adapter is a simple 2-layer fully-connected network inserted to each layer of the language model (LM) to adapt the LM to a specific task. The existing literature has shown that training only the adapter (with the LM fixed) can achieve similar results to fine-tuning the whole LM. In TST, the adapter is also fixed and only the mask to the adapter is trainable to achieve CF prevention (Eq. 1). We add additional details for the adapter in Appendix A.
> >
> > Regarding the gradient, it is evaluated by the current task data (x_t,y_t) and cross-entropy loss L_ce (see Figure 1(A) “forward pass” and its input, output) and Eqs. 2-3). Unlike the mask pool which is different for different layers, there is only one set of virtual gates and it is used for all layers. So its gradient is simply a vector with K values. The backward gradient is then used to compute the importance (see Figure 1(A) “backward pass” and Eqs. 4-5). Note that in the figure we didn’t draw out the “layers” as it gets too crowded.
> >
> > Note that the gradient yields a single mask importance score to the current task (not the “task importance score”). It can achieve this because (1) the gate variable (g^(k) in Eq. 2 is a scalar corresponding to each mask in the pool; (2) the importance score (I^(k)) is computed from the average gradient of all training samples of the task (Eq. 4 and Figure 1(A) “backward pass”); and (3) the importance score (I^(k)) is corresponding to the gradient of each gate variable. Therefore, I^(k) is also a scalar corresponding to each mask. Based on the importance, we select a mask to be used for a task (see Sec. 3.3, and Figure 1(B)). We will be happy to answer any additional questions that you may have about our algorithm.

---

> > > ### Author Response · Authors · 2022-11-18
> > > **Reponse to Reviewer Umdv (Part 3)**
> > >
> > > > My main concern is the lack of novelty .... in some cases the improvement over baselines is inconsistent or relatively small (e.g., SupSup)
> > >
> > > We believe that we have proposed a novel idea. Let us start with what should be achieved for task-incremental learning (TIL). (1) it should achieve both catastrophic forgetting (CF) prevention and knowledge transfer (KT), but relatively limited attention has been paid to KT so far. (2) In order to achieve KT, it is necessary to detect what knowledge is sharable and can be transferred.
> > >
> > > The existing methods (e.g., CAT, CTR) try to detect task similarity and transfer knowledge among similar tasks. However, these methods have two critical weaknesses: (i) They cannot prevent CF during transfer as we discussed in the introduction section and (ii) their similarity detection methods are inaccurate. These are critical issues that hinder the existing methods for KT. We leverage SupSup to learn a sub-network to achieve no interference (CF) in transfer (the original SupSup, by design, cannot do KT). However, it is very challenging to detect task similarity and to know what level of similarity is similar enough to ensure positive transfer. If a wrong similarity threshold is used, CF will be serious.
> > >
> > > Our proposed method has dealt with all these critical challenges for KT and CF in novel ways. We designed a novel method to detect task similarity indirectly without using any threshold via the computation of the importance of each mask (learned for a previous task) to the current task. Although the network pruning community has used the gradient to compute the importance of each parameter, we need to compute the importance of each mask, which is a network with multiple layers and parameters. A single importance score is needed for each mask network so that the importance scores of all masks can be compared. We proposed to use some virtual gates to achieve the purpose (without using any threshold). Thus, we believe that our proposed approach is novel. To our knowledge, no existing method has something close to what we do. If you know these ideas have been proposed somewhere else, we would very much like to know.
> > >
> > > We have talked to some industrial partners and come to the conclusion that the continual learning (CL) community should put some major effort to study KT because without it to improve the results, TIL is hardly useful in applications as one can always build a separate model for each task, which has no CF at all. Although it is more resource-demanding, that is not normally a big issue in practice because practical applications almost always want better results/accuracy.
> > >
> > > Regarding the improvement, it is actually consistent. Among the 5 datasets, 2 of them (ASC and NER) are with similar tasks and the systems need to achieve both CF prevention and KT. 3 of them (SUM, CCD, DRG) are with dissimilar tasks and have little knowledge to share or to transfer. Then, they only need to deal with CF (we have made these clearer in the paper (right after 4.1 (5)). From Table 1, it is clear that TST is the only system that can achieve the above.  We can see the improvement on similar task datasets (ASC and NER) are significant (standard deviation in Appendix E (Tables 7-8)). For the other 3 datasets (SUM, CCD and DRG), as they have very different tasks and thus do not share much knowledge to be transferred, the best that TST can do is not to forget what has been learned. We have made these clearer in the paper (right after 4.1 (5)). We can see TST is similar to ONE (the control baseline that learns each task in a separate network) and SupSup, which indicates TST's effectiveness in forgetting prevention.

---

### Official Review · Reviewer_k1vn · 2022-10-29

**Confidence:** 4
**Correctness:** 3
**Technical Novelty And Significance:** 3
**Empirical Novelty And Significance:** 3
**Recommendation:** 6

**Clarity, Quality, Novelty And Reproducibility:**

Clarity: good
Quality: good
Novelty: incremental
Reproducibility: yes

**Strength And Weaknesses:**

Strengths:
- paper address key questions in continual learning and applied to NLP tasks
- the approach is incremental yet effective
- novel ideas in computing task similarity using mask importance applied to presenting CF and encouraging knowledge transfer, however inspired from existing works
- paper is well motivated and clearly written
- evaluates the proposed approach on 5 different NLP data sets
- paper employs a good set of related baselines to empirical comparisons
- sound quantitate analysis and ablation study

Weaknesses:
- incremental novel steps (limited novelty) yet effective and simple
- unclear / missing experimental setup: how is the data split over time across tasks in Cl setup? Does a different sequence of tasks leads to different performances?
- unclear how the approach address the question: how new entities are learned over time, not seen in the historical tasks? How does the SoftMax adjust and the parameters shared over a sequence of NRE tasks as new entities evolve?

Question:
- See Weaknesses section
- How does the approach scale to unsupervised tasks - such as topic modeling in continual learning paradigm [1]? How does the task similarity be applied on document/text latent representations: topics?
- Include additional existing works such as [2] in continual topic modeling: Unsupervised Continual learning for NLP that overcomes CF via both regularization approach and data replay and computes task similarity using topics - that represents data documents [1] and perform KT.
It is interesting to see how the tasks similarity approaches scales to unsupervised continual learning (close to real world).


References:
[1] Topic modeling using topics from many domains, lifelong learning and big data. ICML 2014.
[2] Neural Topic Modeling with Continual Lifelong Learning. ICML 2020.



**Summary Of The Paper:**

- This works address two majors challenges in continual learning: catastrophic forgetting  and knowledge transfer (KT) across tasks applied to NLP tasks.
- In doing so, the approach is based on sub-networks, and computes task similaity via gradients and mask importance where catastrophic forgetting is controlled using a different mask while knowledge transfer across tasks is encourage using the same mask across tasks.
- The proposed approach is applied to several NLP tasks using 5 datasets, and evalued against exhaustive baselines. The paper show improved performance in terms of controlling catastrophic forgetting  and performing knowledge transfer (KT) across tasks
- The paper is clearly written and well motivated


**Summary Of The Review:**

Please see above.

---

> ### Author Response · Authors · 2022-11-18
> **Response to Reviewer k1vn**
>
> > incremental novel steps (limited novelty) yet effective and simple
>
> We believe that we have proposed a novel idea. Let us start with what should be achieved for task continual learning (Task-CL). (1) it should achieve both catastrophic forgetting (CF) prevention and knowledge transfer (KT), but relatively limited attention has been paid to KT so far. (2) In order to achieve KT, it is necessary to detect what knowledge is sharable and can be transferred.
>
> The existing methods (e.g., CAT, CTR) try to detect task similarity and transfer knowledge among similar tasks. However, these methods have two critical weaknesses: (i) They cannot prevent CF during transfer as we discussed in the introduction section and (ii) their similarity detection methods are inaccurate. These are critical issues that hinder the existing methods for KT. We leverage SupSup to learn a sub-network to achieve no interference (CF) in transfer (the original SupSup, by design, cannot do KT). However, it is very challenging to detect task similarity and to know what level of similarity is similar enough to ensure positive transfer. If a wrong similarity threshold is used, CF will be serious.
>
> Our proposed method has dealt with all these critical challenges for KT and CF in novel ways. We designed a novel method to detect task similarity indirectly without using any threshold via the computation of the importance of each mask (learned for a previous task) to the current task. Although the network pruning community has used the gradient to compute the importance of each parameter, we need to compute the importance of each mask, which is a network with multiple layers and parameters. A single importance score is needed for each mask network so that the importance scores of all masks can be compared. We proposed to use some virtual gates to achieve the purpose (without using any threshold). Thus, we believe that our proposed approach is novel. To our knowledge, no existing method has something close to what we do. If you know these ideas have been proposed somewhere else, we would very much like to know.
>
> We have talked to some industrial partners and come to the conclusion that the continual learning (CL) community should put some major effort to study KT because without it to improve the results, Task-CL is hardly useful in applications as one can always build a separate model for each task, which has no CF at all. Although it is more resource-demanding, that is not normally a big issue in practice because practical applications almost always want better results/accuracy.
>
> > how is the data split over time across tasks in Cl setup? Does a different sequence of tasks leads to different performances?
>
> The tasks are randomly ordered (see the first paragraph in Sec. 4.2). Yes, a different sequence/ordering of tasks can lead to a different performance. That is why we follow the standard CL evaluation (e.g., [a,b]) and average over 5 random task sequences. In Table 1, we can see TST outperforms the baselines on average regardless of the task ordering.
> To make the results more convincing, we give the results for all 5 different sequences in Appendix F (Table 9). From the results for individual sequences, we can observe that the order of tasks indeed affects the results but not by much. In summary, we believe the average result over random sequences in Table 1 shows the effectiveness of TST.
>
> [a]: Ke et al., Achieving Forgetting Prevention and Knowledge Transfer in Continual Learning, NeurIPS 2021
>
> [b]: Zhao et al., Prompt Conditioned VAE: Enhancing Generative Replay for Lifelong Learning in Task-Oriented Dialogue, EMNLP 2022
>
> > how new entities are learned over time? How does the SoftMax adjust?
>
> Please note that we are working in the task continual learning (Task-CL) setting (first paragraph in Sec. 1), which means the task ID is known in both training and testing. For classification tasks (ASC, CCD and NER), we adopt the multi-head setting, i.e., each task has its own head, so there is no need to adjust the softmax. In NER (like other classification experiments), when a new task (which contains its own set of entities) comes, it cannot access any learned data from previous tasks. It trains the shared body (in our case, the masks) and its own token-level classification head. In testing, we use the corresponding mask (which may be shared with some other tasks) and its corresponding classification head to do prediction
>
> > How does the approach scale to unsupervised tasks? Include additional existing works such as [2] in continual topic modeling
>
> In this paper, we focus on supervised tasks. The current technique is not suitable for unsupervised tasks because we need the loss for each input sample in Eq. 4 to be computed in the forward pass, which is not possible for unsupervised learning. We will investigate how to adapt TST for unsupervised tasks in our future work. We have cited [2] and made this clearer in the paper (footnote 3).

---

### Official Review · Reviewer_iK2Y · 2022-10-30

**Confidence:** 2
**Correctness:** 2
**Technical Novelty And Significance:** 2
**Empirical Novelty And Significance:** 2
**Recommendation:** 3

**Clarity, Quality, Novelty And Reproducibility:**

**Clarity and Quality**: The analysis are not clearly demonstrated.

**Novelty**: The novelty of the proposed method is enough.

**Reproducibility**: The author upload code for reproduction.

**Strength And Weaknesses:**

**Strength**

* The motivation behind the design of TST (i.e., assigning sub-network masks based on task similarity) is promising.
* The design of importance score (Equ. 2-5) and the choosing strategy (Equ. 7) is interesting. Moreover, the analysis results in Table 3 and 4 show the effectiveness of these designs.

**Weakness**

* The impt mask defined in Equ. 2 and used in Equ. 3 is a little weird. It seems that the elements in impt mask are not 0/1 and could be extremely larger than 1. Therefore, multiplying this mask to the original weight matrix may strongly affect the functionality of the original modular (i.e., the adapter), and the gradients (which will be used to calculate the importance score) can also be influenced.
* Lack of further analysis of the mechanism behind TST.  I suppose the most important and interesting part in this work is the designs in Equ. 2-7. I agree that these designs sounds good, but I feel it is too amazing for a neural model to perform in a so interpretable way.
This is my major concerns to this work. I list several concrete questions as follows and will change my score according to your response.

**Questions**

* Q1: **Did you cherry pick the visualization in Figure 3?** Your analysis in 'Effectiveness of task similarity detection' tried to demonstrate that the actions (or called the choices of masks) in TST are in line with the intuition that the similar tasks share the mask while dissimilar tasks choose different masks.
Therefore, I'm wondering if these analysis are from the cherry picked model or general ones.
For instance, for the two tasks 'icsi' and 'ami' in SUM, when the random seed and task sequence order are changed, can TST still find them similar?
* Q2: **Does Equ. 7 work correctly at the beginning of the task sequences?** Since masks are randomly initialized, the importance scores calculated in the first several tasks could be hugely affected by this randomness, and the comparison in Equ. 7 might also be influenced. Moreover, I notice that for (a) (b) (e) in Figure 3, TST found most similar tasks at the beginning of the task sequences. Therefore, I'm wondering whether these 'similar tasks' are superficial correlations according to their order in task sequences.

**Minor Questions**

* How are the adapter used in TST trained? Since the paper claims that only the mask is trained, I wonder if the adapter is just randomly initialized without further training?

**Summary Of The Paper:**

This work introduces a new continual learning method TST.
TST considers both preventing catastrophic forgetting (CF) and encouraging knowledge transfer (KT).
To prevent CF, TST directly use one existing CL method called SupSup.
To encourage KT, TST borrows the idea of gradient-based importance score and applies this score to assign sub-network masks for sequential tasks.
Experimental results show that TST meets its motivation.
Concretely, it improves the CL performance on similar tasks and keeps low forgetting rates.


**Summary Of The Review:**

**Post-revision update:**
The response make me further worried about the relation between the 'performance improvement' and the story-telling of 'task similarity detection'.
The insight about 'task similarity detection' is not well-supported.
Thus, I decrease the overall score.


I have some concerns for the effectiveness of the proposed method and therefore give a borderline score.
I list several questions and will change my score according to the response.

---

> ### Author Response · Authors · 2022-11-18
> **Reponse to Reviewer iK2Y**
>
> > ... the elements in impt mask are not 0/1 and could be extremely larger than 1. Therefore, multiplying this mask to the original weight matrix may strongly affect the functionality of the original modular (i.e., the adapter), and the gradients (which will be used to calculate the importance score) can also be influenced.
>
> We believe that there is a misunderstanding here. First, we want to clarify that the mask ($M^{(k)}$) in Eq. 2 is always binary {0, 1} and thus can be used to learn a sub-network (more details about mask training to facilitate the understanding are given in Appendix B). Regarding the importance mask ($M^{impt}$), since the gate variable ($g^{(k)}$) always has the value of 1, the importance mask indeed can be larger than 1. However, please note that we are not updating the importance mask or gate variable when computing the importance (see the paragraph before Eq. 2). Instead, we only use the gradient to compute the importance using Eqs. 4-5 (but not use the gradient to update any parameter). What we really train is only the final selected mask with Eq. 1 (Sec. 3.4)
>
> > Q1: Did you cherry pick the visualization in Figure 3? Q2: Does Equ. 7 work correctly at the beginning of the task sequences?
>
> We want to first clarify that the task orders are randomly generated. We show the random sequences for all datasets in Appendix I (Table 10). We also give the similarity detection results of all 5 sequences and 5 random seeds for all datasets in Appendix I (Table 11).
>
> Q1: The visualization of Figure 3 is taken from the results of the first sequence of each dataset (order 1). We didn’t do any cherry-picking. But we understand your concern. As stated in the Sec. 4.2, the order of tasks matters. That is why we reported the average results over 5 random sequences (orders of tasks) in Table 1 (this is also the standard practice for CL, e.g. [a,b]). We can see TST outperforms the baselines regardless of the order of tasks.
>
> Q2 and Q1: From Appendix I (Table 11), we observe that
> - The similarity detection results in different orders and different seeds have some differences but they are not large. It is reasonable to have some differences as the learning process of the tasks is quite dynamic.
> - The effect of random seeds and orders is different across different datasets. For example, in SUM, similarity detection results for different seeds are the same, probably because each task in SUM is a long document and has more information and easier to detect the similarity. In other datasets, we can see different orders and seeds have some differences, indicating that it is more difficult to detect similarity for them. This may be because the tasks in these datasets are sentence-based text classification (or extraction or generation) and the information contained in a sentence is limited. Thus, the detection results have a larger variance. ASC has the most difference, which is reasonable because its tasks (sentiment classification) are all similar to different extents.
>
> Note that as we discussed in the paper (“Effectiveness of task similarity detection” in Sec. 4.2), the task similarity is fuzzy and it is hard to say which task similarity is actually correct or wrong because there is no ground truth.
>
> To answer your questions, we can see from Appendix I (Tables 10-11) that TST does not always give the beginning tasks as similar. We can also see TST can still find “icsi” and “ami” similar in different random seeds and different orders (except order 5). These show that 'similar tasks' are not superficial correlations.
>
> Regarding the randomness of the first several tasks, it is not an issue because what we care about in Eq. 7 is whether a mask should be shared. The candidate shared mask must be a mask that has been trained and thus the randomness does not affect much. For the mask that is not shared, the system can randomly select a mask from the pool.
>
> In summary, we believe the average over random sequences in Table 1 can show us the effectiveness of TST.
>
> [a]: Ke et al., Achieving Forgetting Prevention and Knowledge Transfer in Continual Learning, NeurIPS 2021
>
> [b]: Zhao et al., Prompt Conditioned VAE: Enhancing Generative Replay for Lifelong Learning in Task-Oriented Dialogue, EMNLP 2022
>
> > How are the adapter used in TST trained? Since the paper claims that only the mask is trained, I wonder if the adapter is just randomly initialized without further training?
>
> Yes. The adapter is randomly initialized and fixed without training. We have made it clearer in the writing (the paragraph before Sec. 3.1).

---

> > ### Comment · Reviewer_iK2Y · 2022-11-19
> > **Thanks for your response**
> >
> > The detailed results shown in Table 11 make me upset about this 'task similarity detection method', since it seems that the method's judgement on task similarity is untrustworthy.
> >
> > Concretely, in your main text, you say that the method can 'clearly outperforms all baselines for tasks with shared knowledge (ASC and NER)', and for other tasks (SUM, CCD and DRG), the performance is not improved.
> > But the results in Table 11 show that the method can consistently judge that certain tasks in SUM, CCD and DRG are similar, but cannot give a consistent judgement on ASC and NER.
> > For instanse, you say that 'all tasks of DRG are dissimilar', but your 'task similarity detection method' consistently find that two tasks *attraction* and *train* in DRG are similar.
> >
> > It seems that **when the method can concretely tell the similarity among tasks, the method do not bring improvement; when it is confusing, it has benefits**.
> > I'm now more wondering about **whether this 'task similarity detection method' really bring improvements from 'task similarity detection'**.
> >
> > > ... each task in SUM is a long document and has more information ... the information contained in a sentence is limited ...
> >
> > You introduce a new factor 'amount of information' that could affect the proposed method.
> > As you described in your response, the influence from this factor is considerable, but it is not mentioned in your main text, and is not experimentally analyzed.
> >
> > > ... the task similarity is fuzzy and it is hard to say which task similarity is actually correct or wrong ...
> >
> > I understand.
> > I do not expect that one method can perfectly find the task similarity.
> > My expectation is much lower: when the method **can** find the task similarity, it will improve the performance; when it **cannot**, it will not bring benefits.
> > However, the results in Table 11 are the exact opposite of what I expected.
> >
> > In summary, the main contribution of this work is the proposed 'task similarity detection method' and the performance improvement from this method.
> > But the author could not clearly demonstrate that the improvement is from 'task similarity detection'.
> > An influential factor, 'amount of information', is introduced in response, which is not well analyzed in the paper, and further indicates that the improvement could be strongly influenced by other factors beyond 'task similarity'.
> > Thus, although the method can bring improvements in some certain tasks, the experimental results do not sufficiently support the story-telling about 'task similarity detection'.

---

> > > ### Author Response · Authors · 2022-11-20
> > > **Response to iK2Y’s response**
> > >
> > > Thank you for your very careful analysis. We are happy to answer your question to clear your doubts. As we have experienced, what happens in knowledge transfer is very subtle and can be hard to understand. In our work, we did a lot of analyses of results ourselves in terms of both similarity detection and improvements to make sure what we are doing is right and consistent. Sorry for the long explanation due to the subtlety.
> > > > When the method can concretely tell the similarity among tasks, the method does not bring improvement; when it is confusing, it has benefits.
> > >
> > > Let’s first look at similar task datasets (ASC and NER). Take ASC as an example. As we know, sentiment classification tasks are somewhat similar. To be more precise, we want to emphasize that we do not really know how many and which tasks are really similar but the majority of them should be similar. As a result, it doesn’t matter what tasks are detected as similar, it is likely to bring improvements.
> > >
> > > Regarding the reason for the inconsistent task similarity detection in different orderings of tasks, we were puzzled in the early stage of our work too, but then we found that it is quite natural due to the dynamics of the sequential learning process. Let us use an example to explain. Let tasks 1, 2 and 3 be all somewhat similar. Say the task order is 1, 2, 3. When learning 2, we find that 1 is similar to 2, and 2 will be learned in the mask for task 1. When learning 3, we may not find 3 to be similar to 1 or 2 because we do not have 1 or 2 individually any more, but the combination of 1 and 2 as they use/share the same mask. So different task orderings may result in different similarities.
> > >
> > > Now let’s see the “dissimilar” tasks. We use double quotes here because we feel they are dissimilar, but there may be hidden similarities. Take SUM as an example. We feel the summarization tasks are different in both domain and the output summary. However, we do not know whether there are any hidden similarities. TST needs to be very careful not to detect dissimilar as similar because it will bring serious forgetting. We can see TST gives consistent detection results, and there is no forgetting, which indicates that TST’s similarity detection is trustworthy. The bottomline is that we need to be conservative, i.e., ensuring no forgetting first and then encouraging knowledge transfer.
> > >
> > > A further question is, if TST can detect some similar tasks in SUM and they look reasonable (e.g., icsi and ami), why don’t the detected similar tasks improve in performance? If we use SupSup as control (as our method is based on SupSup), which trains a separate mask for each task, we can see TST is almost the same as SupSup for SUM, CCD and DRG. This confirms that the detected similar tasks have no forgetting and they also do not help each other in the three dissimilar datasets. This is actually possible just like transfer learning (we worked on transfer learning before) because there is no guarantee that similarity can result in improvements. Of course, it is also possible that a better transfer technique is needed.
> > >
> > > In conclusion, we think that your expectation *when the method can find the task similarity, it will improve the performance; when it cannot, it will not bring benefits* could be changed to **when the method can find similar tasks, it should bring improvement but may not in some cases; when it cannot, it should avoid forgetting (no improvements)** (benefits means improvement if we understand you correctly). We hope this explains why the task similarity detection is trustworthy and how it brings improvement.
> > > > An influential factor, 'amount of information', is introduced in response, which is not well analyzed in the paper, and further indicates that the improvement could be strongly influenced by other factors beyond 'task similarity'.
> > >
> > > We use the term “information” in a loose sense, not meant to be the information in the information theory. What we meant was that the information in a long document (e.g., in summarization tasks) is more than that in a sentence (e.g., in ASC sentence-based sentiment classification task), which may make similarity detection easier. We did not put this in the paper as it is not a main point, but just the nature of the input of different problems.
> > >
> > > We hope you can reconsider your decision.

---

> ### Author Response · Authors · 2022-11-28
> **To Reviewer iK2Y: a gentle reminder of our new response**
>
> Dear Reviewer iK2Y,
>
> We are writing to check whether you have further questions on our response. We understand your expectation and gave a more appropriate one in our further response. We also show how the detailed results in Table 11 are reasonable and support the fact that the similarity detection is effective. If you have any further questions, we are more than happy to discuss. We are also willing to give more information or perform more experiments to address your concerns.
>
> Thanks,
> Authors

---

> > ### Comment · Reviewer_iK2Y · 2022-12-06
> > **Thanks for your further explanation!**
> >
> > It seems that there are too many trival factors that could influence the model behavior and its effectiveness.
> > Since I'm not sure whether my expectation is too high to achieve, I reduce my confidence score.

---

### Official Review · Reviewer_ktyy · 2022-11-04

**Confidence:** 5
**Correctness:** 3
**Technical Novelty And Significance:** 3
**Empirical Novelty And Significance:** 2
**Recommendation:** 5

**Clarity, Quality, Novelty And Reproducibility:**

The paper is well-written and easy to follow. Moreover, the proposed solution is grounded in recent works on parameter-efficient transfer learning-based approaches. Also, the proposed mask importance measure is built on existing pruning literature and is successfully applied to the continual learning problem. The paper also provides its code thereby convincing the reader of the reproducibility of their experiments.

Typos in Eq(4) and Eq(5): should it be $\nabla_{g_{(k)}}^n$?


**Strength And Weaknesses:**

Strengths:

1. The paper attempts to solve an important problem of task continual learning with a specific focus on reducing forgetting and enabling forward transfer objectives. These objectives constitute important desiderata for realistic continual learning systems.

2. The proposed TST approach employs parameter-efficient adapter modules to learn sub-networks for different tasks, enabling their approach to scale to a large number of tasks. Also, the proposed task similarity detection method based on gradients is easy to compute and seems effective.

3. The paper conducts a rigorous set of experiments spanning different NLP tasks and compares their performance with several relevant baselines.

Weakness:

1. Although the paper conducts several experiments, there are some open questions regarding the statistical significance of the results provided in Table 1. Specifically, comparing the results with SupSup, it is unclear whether TST is the clear winner (see SUM, CCD, DRG). As the paper reports, the average results over 5 random task orderings, please consider reporting significant test results.

2. Apart from alleviating forgetting of the previous task, the paper claims that the TST approach also encourages knowledge transfer. Results presented in Table 1 do not provide any convincing evidence for the same. The paper should consider reporting the learning accuracy as mentioned in [1]. For example, to understand the knowledge transfer from Tasks 1 and 2  while learning Task 3, one should consider reporting average performance for tasks after learning on their datasets. Also, if the paper argues that there is positive backward transfer in that case to compute the forgetting, the drop in the performance should be from the max performance for previous tasks and not just right after Task i. Please see the definitions of forgetting and learning accuracy in Eq(1) in [1] for reference.

References:

[1] Mehta, Sanket Vaibhav, Darshan Patil, Sarath Chandar, and Emma Strubell. "An empirical investigation of the role of pre-training in lifelong learning." arXiv preprint arXiv:2112.09153 (2021).


**Summary Of The Paper:**

The paper studies the problem of task continual learning wherein new tasks are continuously learned in an incremental fashion. Specifically, the paper aims to achieve two key objectives for a successful continual learning system: preventing catastrophic forgetting of the previous tasks and enabling knowledge transfer to new tasks. To achieve this, the paper proposes the TST approach, Task-continual learning based on Sub-networks and Task similarity. The main idea is to start with a mask pool, then select the relevant mask for the given task with gradient-based importance measure, and fine-tune it with the current task data. With the proper selection of the relevant mask, one can alleviate the interference from unrelated tasks and simultaneously enable backward/ forward transfer to related/ current tasks. The paper conducts an extensive set of experiments on a wide range of NLP tasks - sentence-level classification, token-level classification, and generation and compares the proposed TST approach with relevant baselines. In summary, the paper shows that TST improves the overall accuracy of the system after sequentially training on different tasks, reduces forgetting, and in some cases (ASC and NER) even enables positive backward transfer.

**Summary Of The Review:**

Overall, I rate this paper marginally below the acceptance threshold. It is interesting to see the efficacy of the proposed approach for task continual learning. However, with the currently reported numbers, it is unclear how significant the improvements are over considered baselines. The addition of the statistical significance test results would be more convincing and I would be happy to increase my score during the discussion phase.

---

> ### Author Response · Authors · 2022-11-18
> **Response to Reviewer ktyy**
>
> > ...there are some open questions regarding the statistical significance of the results provided in Table 1. Specifically, comparing the results with SupSup, it is unclear whether TST is the clear winner (see SUM, CCD, DRG)''
>
> We actually reported the standard deviation in Appendix E (Tables 7-8) and mentioned that in the caption of Table 1. You can see the improvement on similar datasets (ASC and NER) are significant. Regarding the 3 datasets you mentioned, they have very different tasks and thus do not share much knowledge to be transferred and the best that TST can do is not to forget what has been learned. We have made these clearer in the paper (Sec. 4.1). We can see TST is similar to ONE (a control baseline that learns each task separately) and SupSup, which indicates TST’s effectiveness in forgetting prevention.
>
> > ...The paper should consider reporting the learning accuracy as mentioned in [1]. .... Also,..., the drop in the performance should be from the max performance for previous tasks and not just right after Task i.
>
> In our experiments, knowledge transfer can be observed in several ways: (1) using ONE as control, we can see TST outperforms ONE in two datasets as they consist of similar tasks and thus have shared knowledge to transfer (details in Sec. 4.2 (1)). The other three datasets all consist of dissimilar tasks and thus have little shared knowledge to transfer in order to do better; (2) using the forgetting rate, we can see there are negative forgetting rates, which indicate positive backward knowledge transfer (detailed in Sec. 4.2 (knowledge transfer and forgetting prevention)); Also see the forward knowledge transfer below.
>
> Regarding the alternative metrics in [1], we did not include them because we think (1) and (2) above are already strong indicators. We further report the learning accuracy (named forward performance (TST (forward)) to be more intuitive) in Table 1. Clearly, it is better than ONE, which indicates effective forward knowledge transfer.
>
> Regarding the “drop of performance”, we believe you mean the “F/Forgetting” (Eq. 1 in [1]). We want to clarify that we follow the forgetting rate in the literature, e.g., [a, b]. It is hard to say which metric for the forgetting rate is correct. Since both metrics measure everything from the standing point of the end of continual learning, i.e., after all tasks are learned, we believe our metric from [a,b] is more reasonable than that in [1] that takes max. We use the following 4 examples to illustrate.
> - Ex 1: if task 1 archives the accuracy 0.5 right after its training, it achieves 0.4 after task 2, and 0.3 after task 3 (final task). In this case, both metrics give the same forgetting 0.2.
> - Ex 2: if task 1 archives the accuracy 0.5 right after its training, it achieves 0.8 after task 2 and 0.82 after task 3 (final task). If we take max [1], then the forgetting is -0.02, but our method will give -0.32. In this case, our method is more reasonable because it precisely shows how much backward knowledge transfer (negative value here means backward transfer) has been achieved for task 1 after task 2 and task 3 are learned since both metrics evaluate from the same reference point, i.e., after all tasks are learned (the last task is t in both measures, i.e., task 3 in this example).
> - Ex 3: If task 1 archives the accuracy 0.5 right after its training, 0.8 after task 2 and 0.4 after task 3 (final task), our metric will give the forgetting 0.1 while Eq. 1 in [1] will give 0.4. This case is more debatable because [1] catches the worst forgetting. But since both measures evaluate after all tasks are learned (the reference point is when the last task is learned), again we believe that our method is more reasonable. We understand if you dispute this. Please see Ex 4.
> - Ex 4: If task 1 archives the accuracy 0.5 right after its training, 0.1 after task 2 and 0.4 after task 3 (final task), both metrics will give the forgetting 0.1. In this case, [1] does not catch the worst forgetting of 0.4 (0.5-0.1) in the process. Again, if we agree that both metrics evaluate from the reference point of when the last task is learned, then both metrics are fine in this case.
> We understand if you may not completely agree with us. Perhaps a better measure should be designed. We have cited [1] and discussed the above in Appendix G. We could not include it in the main text as it is too long.
>
> [a]: Liu et al., Mnemonics training: Multi-class incremental learning without forgetting. CVPR 2020.
>
> [b]: Ke et al., Achieving Forgetting Prevention and Knowledge Transfer in Continual Learning, NeurIPS 2021
>
> > Typos in Eq(4) and Eq(5): should it be $\nabla^n_{g^{(k)}}$
>
> Response: Sorry, this is a typo and we have revised it.

---

### Decision · Program_Chairs · 2023-01-20

**Decision:**

Reject

**Justification For Why Not Higher Score:**

Given the reviews and the interaction with the authors, I felt that the paper could be improved if the feedback from the reviewers are used to improve the paper.

**Justification For Why Not Lower Score:**

N/A

**Metareview: Summary, Strengths And Weaknesses:**

The paper presents a new method called TST:  Task-CL based on Sub-networks and Task similarity, which is a continual learning method that is based on subnetworks and task similarity.  The authors claim that this method overcomes catastrophic forgetting while also helping in knowledge transfer.

Strengths:  The problem space is very important, the methods are elegant and seem to be generally quite easy to use, and the reviewers like the experimental setup and it is well written.

Weaknesses: Reviewers have questions about the significance of the results, questions about how convincing the knowledge transfer experiments are.  Reviewer iK2Y have concerns about how justified the purported improvements are due to the mentioned insights of the TST formulation.